# Heterogeneous side effects of cortical inactivation in behaving animals

**Ariana R Andrei[1], Samantha Debes[1], Mircea Chelaru[1], Xiaoqin Liu[1], Elsa Rodarte[2], John L Spudich[3], Roger Janz[1], Valentin Dragoi[1]\***

[1]Department of Neurobiology and Anatomy, McGovern Medical School, University of Texas, Houston, United States; [2]Department of Neurology, McGovern Medical School, University of Texas, Houston, United States; [3]Center for Membrane Biology, Department of Biochemistry and Molecular Biology, McGovern Medical School, University of Texas, Houston, United States

**Abstract** Cortical inactivation represents a key causal manipulation allowing the study of cortical circuits and their impact on behavior. A key assumption in inactivation studies is that the neurons in the target area become silent while the surrounding cortical tissue is only negligibly impacted. However, individual neurons are embedded in complex local circuits composed of excitatory and inhibitory cells with connections extending hundreds of microns. This raises the possibility that silencing one part of the network could induce complex, unpredictable activity changes in neurons outside the targeted inactivation zone. These off-target side effects can potentially complicate interpretations of inactivation manipulations, especially when they are related to changes in behavior. Here, we demonstrate that optogenetic inactivation of glutamatergic neurons in the superficial layers of monkey primary visual cortex (V1) induces robust suppression at the light-targeted site, but destabilizes stimulus responses in the neighboring, untargeted network. We identified four types of stimulus-evoked neuronal responses within a cortical column, ranging from full suppression to facilitation, and a mixture of both. Mixed responses were most prominent in middle and deep cortical layers. These results demonstrate that response modulation driven by lateral network connectivity is diversely implemented throughout a cortical column. Importantly, consistent behavioral changes induced by optogenetic inactivation were only achieved when cumulative network activity was homogeneously suppressed. Therefore, careful consideration of the full range of network changes outside the inactivated cortical region is required, as heterogeneous side effects can confound interpretation of inactivation experiments.

**\*For correspondence:**
Valentin.Dragoi@uth.tmc.edu

**Competing interest:** The authors declare that no competing interests exist.

## Introduction

Determining causal relationships between neuronal circuits and behavior represents a fundamental goal of systems neuroscience. Establishing causality, and not just correlation, is difficult and is typically done by externally modulating neural activity, that is, using optogenetics, thermal cooling, or pharmacological agents, and observing the effects on behavior. In a typical cortical inactivation experiment, a particular neural population is silenced, and if a behavioral impairment is observed, that specific population is believed to be causally involved in modulating that specific behavior. Cortical neurons, however, are embedded in densely interconnected local networks spanning hundreds of microns (*Douglas and Martin, 2004*; *Hirsch and Martinez, 2006*; *Stettler et al., 2002*). These local connections are crucial for contextually modulating neural responses, and they underlie canonical cortical computations such as divisive normalization (*Carandini and Heeger, 2011*) and surround suppression (*Adesnik et al., 2012*; *Angelucci et al., 2017*; *Trott and Born, 2015*). It is unknown how focal suppression influences activity in the local network (*Figure 1A*).

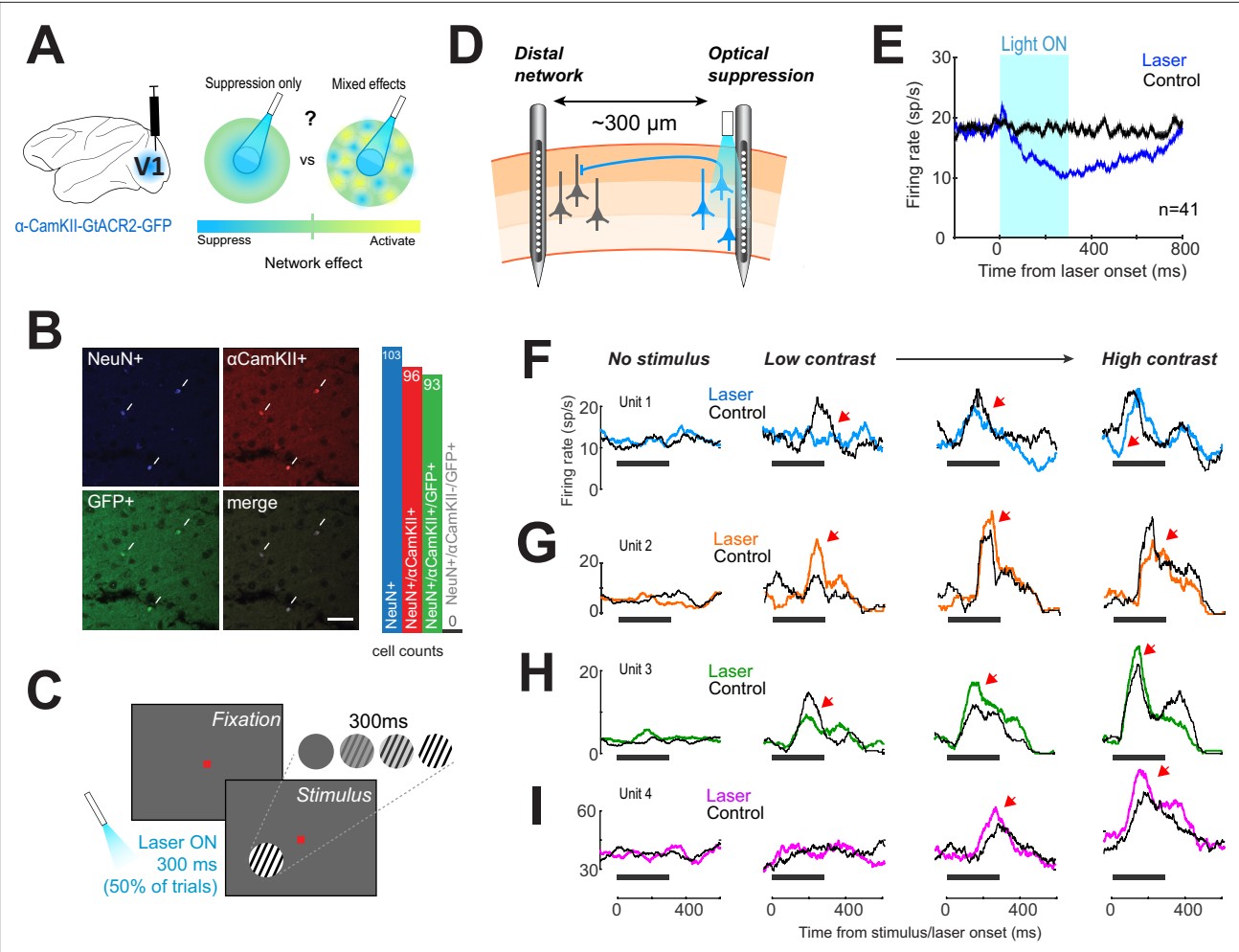

**Figure 1.** Optogenetic suppression yields heterogeneous responses in the distal network. (**A**) Left, glutamatergic neurons were targeted using a lentivirus construct, containing the gene for *Gt*ACR2, under the control of an α-CamKII promoter. Right, optogenetic suppression of a focal neural population may result in a spread of suppression across the network ('Suppression only') or in a heterogeneous response in the network ('Mixed effects'). (**B**) Immunohistochemical analysis performed on biopsied tissue from one monkey, after experiments were complete, confirmed that expression of green fluorescent protein (GFP)-tagged *Gt*ACR2 was confined exclusively to excitatory neurons. White arrowheads indicate neurons immuno-positive for NeuN (pan-neuronal marker, top left panel), α-CamKII (glutamatergic neuron marker, top right panel), and GFP (*Gt*ACR2 marker, bottom left panel). Right, bar graph shows counts for immuno-positive cells counted across five separate sections. No inhibitory neurons (NeuN+/CamKII-) were found to be positive for GFP. Scale bar is 25 μm. (**C**) Animals viewed 300 -ms contrast-varying oriented gratings on a computer monitor. Optogenetic suppression was present on 50 % of trials in a randomly interleaved manner. Light duration was 300 ms and was synchronized with stimulus presentation. (**D**) Electrophysiological recordings using laminar electrodes were made either proximal to the fiber optic location or at distal sites ~300 μm away. (**E**) Neural responses near the light source. Population responses (n = 48) with (blue trace) and without (black trace) laser activation of *Gt*ACR2, while animals fixated. Horizontal black bar shows time of light on. Error envelopes show sem. (**F–I**) Examples of neural responses recorded distal from the fiber optic, across four visual stimulus conditions (columns). Columns are arranged with increasing stimulus contrast (0%, leftmost column, to 100 %, rightmost column). Black traces show the mean firing rate on control trials. Colored traces show mean firing rates on laser trials. Horizontal black line shows timing of the visual stimulus and light on; red arrowheads mark regions of interest.

The online version of this article includes the following figure supplement(s) for figure 1:

**Figure supplement 1.** Example session with simultaneous dual recordings showing direct suppression at the illumination site and heterogeneous responses in the distal network.

In principle, silencing a neural population could likely cause substantial activity disruption in the neighboring connected network. In small-brained animals, such as rodents and fruit flies, it is physically possible to silence an entire cortical area with a small amount of light (*Li et al., 2019*; *Mauss et al., 2017*). However, for deep-brain structures, where light penetration is more difficult, or in animals with larger brains, such as monkeys, complete inactivation of a brain region is often not possible (*Acker*

*et al., 2016*; *Galvan et al., 2017*; *Gerits et al., 2012*). In these latter scenarios, off-target effects of inactivation manipulations must be considered when interpreting behavioral results since nearby neurons are equally likely to contribute to any observed behavioral changes. Such off-target network effects are particularly troublesome for studies that do not record neural activity during light-induced suppression, but rather assume that the net impact on the targeted network is equivalent to the effect of the opsin on an individual cell. However, despite the importance of these issues, the indirect effects of optogenetic silencing on the neighboring neural population activity and on animal's behavior are largely unknown.

Here, we explicitly examined off-target, network-mediated effects through population recordings of neural activity across the cortical layers of primary visual cortex (V1) of behaving monkeys while optogenetically inactivating a local population of excitatory cells using the suppressive opsin *Gt*ACR2 (*Figure 1A*). We found that optogenetic suppression unmasks highly heterogeneous effects in the lateral columnar networks ranging from full suppression to full facilitation. Furthermore, the degree of heterogeneity of indirect network effects following optogenetic inactivation varies as a function of cortical layer. A computational model indicates that these diverse responses can be explained by differences in the strength of local intracortical inputs during distal inactivation, and are likely due to the same circuitry that mediates normalization processes, such as contrast gain control. Finally, we show that behavioral performance is consistently impaired only when the net activities of both the on- and off-target sites are uniformly suppressed, while heterogeneous off-target activity resulted in heterogeneous behavioral changes. Such off-target, network-mediated effects provide a viable explanation for the diversity of results reported in optogenetic studies in non-human primates (*Tremblay et al., 2020*).

## Results

### Optogenetic suppression of V1 responses

To examine whether focal silencing of a population of V1 neurons leads to uniform or heterogeneous off-target responses in the neighboring local network, *Gt*ACR2, a chloride-conducting channelrhodopsin, was expressed in populations of glutamatergic cells (see 'Materials and methods'). *Gt*ACR2 has been shown to be more sensitive to light and produce stronger hyperpolarizing currents than other inhibitory opsins (*Govorunova et al., 2015*). To our knowledge, this is the first usage of this opsin in the monkey brain. To quantify the specificity of gene expression using this novel construct, we developed a novel biopsy technique to extract cortical tissue for immunohistochemical analysis (see 'Materials and methods'). Gene expression was robust and specific to neurons expressing glutamatergic marker α-CamKII (*Figure 1B*), with 97 % of cells positive for α-CamKII also positive for green fluorescent protein (GFP, co-expressed with *Gt*ACR2), and no neurons (NeuN+, a pan-neuronal nuclear antigen marker) that were negative for α-CamKII and positive for GFP (corresponding to inhibitory cells).

Monkeys performed a contrast detection task in which optogenetic suppression was randomly interleaved on 50 % of trials (*Figure 1C*, see 'Materials and methods'). Extracellular recordings were made using multi-contact laminar electrodes and light was emitted by an independently movable fiber optic (*Figure 1D*). We positioned the fiber optic such as to inactivate the superficial layers of V1 (*Andrei et al., 2019*) and measured the amount of direct suppression (the light source was coupled to the recording electrode; *Figure 1D–E*). As expected, light activation reduced firing rates by 24.9% ± 2.1% compared to control trials (mean ± sem, over the entire 300 ms light stimulation interval; 18.1 ± 2.1 sp/s control trials, 11.6 ± 1.4 laser trials, p = 0.0064, Wilcoxon rank sum test, n = 41). The direct suppression was long-lasting, and responses returned to baseline 387.9 ms ± 38.7 ms after light offset (median ± sem), consistent with previously measured kinetics of *Gt*ACR2 (*Govorunova et al., 2015*).

### Heterogeneous network side effects during optogenetic inactivation

Next, we examined the off-target effects of optogenetic inactivation by measuring the changes in neural responses away from the light stimulation site (*Figure 1D*, left). This was done by separating the electrode from the light source by a fixed distance, ~300 μm, around the stimulation site, using a custom grid placed inside the recording chamber. This allowed us to assess whether the nearby columnar networks that were not directly inactivated by light undergo changes in population activity.

We recorded a total of 214 units across 21 sessions (both single- and multi-unit responses were included: 127 units from Monkey 1 and 87 units from Monkey 2), with statistically significant differences in firing rate during laser trials compared to control (no laser) trials at one or more visual contrast conditions (Wilcoxon ranked sum test, p<0.005). Although firing rates of neurons did not change in the blank 0%, contrast condition (*Figure 1F–I*; laser vs control trials), the stimulus-triggered responses were highly heterogeneous. The changes in neural responses varied between robust suppression (*Figure 1F*) and excitation (*Figure 1I*), or mixed effects at various contrasts (*Figure 1G–H*). The lack of suppression in the blank condition indicates these responses cannot be due to direct activation of *GtACR2* at the distal site. We further confirmed that both direct silencing (*Figure 1E*) and heterogeneous responses in the local network (*Figure 1F–I*) are present simultaneously by performing additional recordings using two laminar electrodes, one coupled to the light source, and the other positioned hundreds of microns away (*Figure 1—figure supplement 1A-K*). Further, to assess whether the suppressive responses (such as those in *Figure 1F*) at the distal site were not due to a small amount of scattered light originating from the fiber optic, we measured the suppression latency on laser trials in the presence of visual stimulus (*Figure 1—figure supplement 1L,M*).

As expected from a network-mediated effect, the suppressed cells at the off-target site took significantly longer to exhibit response inactivation (92.9 ± 1.1 ms, mean ± sem) compared to the cells at the direct site (31.2 ± 1.2 ms; p = 0.0007, Wilcoxon ranked sum test). These results demonstrate that local optogenetic suppression results in largely heterogeneous effects across neurons in the vicinity of the directly suppressed cell population. To quantify the diversity of light-induced responses, we first measured the contrast-evoked spike counts for each light-responsive neuron for laser and control trials (see 'Materials and methods'). Next, we fitted the contrast responses with an implementation of the normalization equation tailored for optogenetics experiments (*Nassi et al., 2015*; *Sato et al., 2014*):

$$R\left(c\right) = \frac{R_m * \left(R_o + c^n + P\right)}{\left(C50 + c^n + Q\right)} \tag{1}$$

where R is the modeled response to visual stimulus contrast (*c*), $R_m$ is the maximum firing rate, $R_o$ is the baseline firing rate, *n* is the neuron's sensitivity to contrast, C50 is the semi-saturation constant, and *P* and *Q* represent the extent to which the local network provides activation and divisive suppression, respectively. The normalization model provided good fits for the observed contrast responses (*Figure 2—figure supplement 1B, C*) and was able to capture the diversity of light-modulated neural responses (*Figure 2A*). Across the population, there was a continuum of responses (*Figure 2A and C*), but four basic response patterns clearly emerged following optogenetic inactivation of the nearby network (see the example cells in *Figure 2A* and population responses in *Figure 2B*; see also *Figure 2—figure supplement 1A*).

To categorize individual cell responses into specific types, two converging approaches were used, utilizing both the normalization fits and the raw firing rates across contrasts (*Figure 2—figure supplement 1A*) for all light-responsive neurons. For each neuron, contrast response differences between laser and control conditions were assigned to one of the four possible categories based on visual inspection. For example, cells that showed a decrease in firing rate at most contrasts on laser compared to control trials were tentatively classified as Type 1. Next, the differences in the normalization fits between the laser and control conditions were algorithmically classified into four classes based on the average difference at low contrasts (1 % to c50 of each cell) and at high contrasts (90–100%). These two methods converged on the same result for 88.3 % of the cells. For disagreements, the category that best matched the firing rate pattern was chosen. To validate that these four response categories were representative of the light-evoked firing rate changes of the cell populations and not an artifact of the classification procedure, we used two additional methods. First, we used a clustering algorithm (uniform manifold approximation and projection) (*Meehan et al., 2021*) to group the firing rate differences (laser minus control) across contrasts for all cells (*Figure 2—figure supplement 1D, E*). This method produced four distinct clusters for the actual data (*Figure 2—figure supplement 1D*), but not for the randomly labeled data (*Figure 2—figure supplement 1E*). Second, a linear discriminant classifier (*Figure 2—figure supplement 1F*) correctly classified firing rates into the ascribed response types with an overall accuracy of 74.3% ± 2.6% (mean ± sem across five training

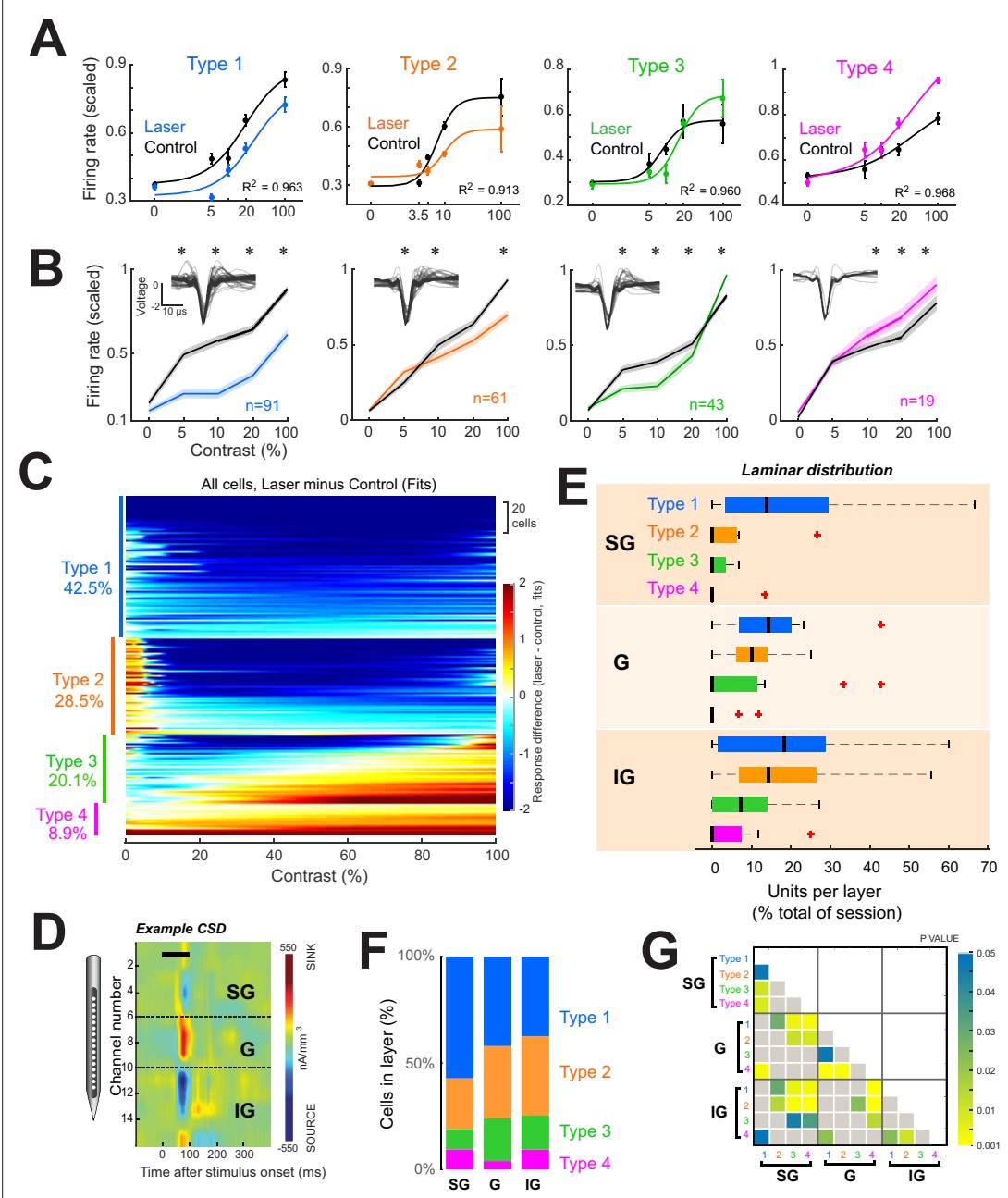

**Figure 2.** Four response motifs observed across cortical layers. (**A**) Representative examples for the four distinct response patterns observed. Points show mean ± standard error of firing rates scaled relative to the maximum across trials. Solid lines show normalization model fits to data points. Black points/lines show basic responses. Colored points/lines show responses during optogenetic suppression of local network. (**B**) Population contrast responses for each response type. *p<0.05, Wilcoxon signed rank test, with false discovery rate correction. Insets show average waveforms from all cells. (**C**) Response types across the population. Heatmap shows differences in firing rate (laser minus control) as predicted by normalization fits for each recorded neuron. Types were classified based on the model fit pattern and ordered based on the mean change in firing rate between laser and control trials across contrasts. (**D**) Example current-sink density estimate for one session, used to assign layer identity. Black horizontal bar shows the time of stimulus presentation. Dashed horizontal black lines show layer boundaries (see 'Materials and methods'). (**E**) Distributions of response types per layer as a percent of all cells recorded in each layer across all sessions. Black vertical line represents the median. Edges of boxes represent the 25th and 75th percentiles. Dashed lines represent the range. Red crosses represent outliers. Inset shows the percent of each response type within each layer. (**F**) Proportion of each response type within each layer. (**G**) Results of statistical comparisons of distributions shown in panel (**E**) (Kruskal-Wallis test, post-hoc Tukey test).

The online version of this article includes the following figure supplement(s) for figure 2:

*Figure 2 continued on next page*

*Figure 2 continued*

**Figure supplement 1.** Response type classification captures firing rate differences.

**Figure supplement 2.** Normalization model fit parameters based on five contrasts are nearly identical to those estimated from nine contrasts.

**Figure supplement 3.** Laminar distribution of response types across sessions with all layers simultaneously recorded.

and validation implementations of the classifier), which was well above chance 33.2% ± 1.9% (p = 1.34 × 10⁻⁶, t-test, two-way, unpaired, n = 5).

Lastly, since our experimental design focused on the low-contrast range of visual stimuli (where behavioral differences are more evident), we verified that the normalization model fit with five predominantly low contrasts produced comparable results to fitting with more numerous, evenly spaced contrasts. To do this, we conducted additional experiments using nine stimulus contrasts (*Figure 2—figure supplement 2A-D*) and recorded the evoked responses of 54 additional units. We fit neuron responses using all nine contrasts and a subsample of five contrasts (with a similar range as the optogenetics experiments). There was a high degree of correlation between the measured c50 and the slope of the fits based on nine vs five contrasts (Pearson *R* = 0.89, p<0.001), demonstrating that the normalization model parameters based on five contrasts were sufficient to capture close to the 'true' contrast response parameters of each cell.

Using this classification strategy in the distal network, we found that a significant fraction of neurons exhibited suppression across all visual stimulus contrasts (labeled Type 1, 42.5 % of all light-modulated cells). However, the majority of light-responsive neurons (57.5%) in the distant network exhibited either mixed effects or facilitation—Type 2: facilitation at low contrasts and suppression at high contrasts (28.5 % of cells); Type 3: suppression at low contrasts and facilitation at high contrasts (20.1 % of cells); and Type 4: facilitation across all contrasts (8.9 % of cells).

## Response heterogeneity is not due to direct optogenetic suppression

In addition to the latency analysis described above (*Figure 1—figure supplement 1L, M*), there are several other reasons why the distal effects we have revealed cannot be due to direct optogenetic suppression. First, direct suppression caused by light activation of *Gt*ACR2 channels has a very characteristic pattern, comprising a strong hyperpolarization (or decreased firing rate) that persists hundreds of milliseconds *after* light offset. This has been demonstrated in vitro (*Govorunova et al., 2015*) and confirmed in our in vivo data (*Figure 1E* and *Figure 1—figure supplement 1C*). These long-lasting responses are only found at recording sites adjacent to the light source, but never at the distal recording sites. Second, the distal recording sites are located well within the area targeted by the virus injections. Injections were spaced across the cortical surface within a rectangular area spanning 1.0 × 0.7 mm (four horizontally spaced columns, with 1 µl × 5 vertically spaced injections in each column; see 'Materials and methods'). Given that 1 µl of a lentiviral suspension has been shown to transfect cells within 1 mm³ (*Han et al., 2009*), glutamatergic neurons at the distal site would also be expected to express *Gt*ACR2 and should be suppressed if exposed to light. This is particularly true since *Gt*ACR2 is much more sensitive to light than traditional channelrhodopsins (*Govorunova et al., 2015*). This also means that the distal heterogeneous effects are not due to retrograde transport of the virus, since the neurons at the distal site are located within the injection boundaries. Thus, if light traveled to these distal sites, the transfected cells should exhibit characteristic *Gt*ACR2-mediated suppression, and this effect would be most apparent in the absence of a visual stimulus. However, we never observed cells exhibiting the characteristic long-lasting suppression at distal recording sites. Further, neural responses at the distal sites had no significant light-induced modulation in the absence of the visual stimulation (*Figure 2B*, *Figure 2—figure supplement 1A*; the 0-contrast condition). This latter finding is consistent with the network-based modulation activated by the visual stimulus, but cannot be attributed to direct activation of *Gt*ACR2 channels by stray light. Lastly, blue light (473 nm wavelength) is known to scatter rapidly in brain tissue (*Diester et al., 2011*), with virtually no light penetrating laterally beyond the collimated area (*Li et al., 2019*). We confirmed the absence of direct light activation at the distal site by performing experiments with simultaneous recordings at both the light-application site and a second distal site (*Figure 1—figure supplement 1*). In these experiments, we confirmed that direct, long-lasting suppression, regardless of stimulus contrast, was only found along the electrode adjacent to the light source (*Figure 1—figure supplement 1C*, n =

12). Despite being located within the injection-targeted area, not such characteristic, direct-light-mediated responses were found at the distal sites. The responses along the distal probe were only of the stimulus-dependent, heterogeneous variety (*Figure 1—figure supplement 1B*, n = 5; n = 3 Type 2 and n = 2 Type 3 response motifs).

## Laminar distribution of response types

We next asked whether there is a laminar specificity to the different response type patterns induced by the suppression of the nearby cell population in the superficial layers. In principle, local intracortical connections could be involved in transmitting the direct optogenetic suppressive effects observed at the light stimulation site to nearby cortical columns. However, since local connections are distance-dependent, the suppression of cortical responses at the light stimulation site could result in direct suppression in the superficial layers of adjacent microcolumns, whereas the effects at larger depths could be mixed.

To measure these effects, we identified V1 layers using a standard current-sink density (CSD) analysis (*Hansen et al., 2012*) based on the local field potentials recorded in response to a large (5 °) full-contrast dynamic visual stimulus presented for 1600 ms (*Figure 2D*, see 'Materials and methods'). A clear laminar profile could be identified in 15/21 sessions. Allocating units in these sessions to their respective layers (n = 24, 51, 80 for supragranular (SG), granular (G), and infragranular (IG), respectively) reveals different proportions of response types across layers (*Figure 2E–G*). Specifically, Type 1 responses dominate in SG, while all types are evenly distributed in G and IG (*Figure 2E* and *Figure 2—figure supplement 3*). Type 1 (purely suppressed) units were evenly distributed throughout

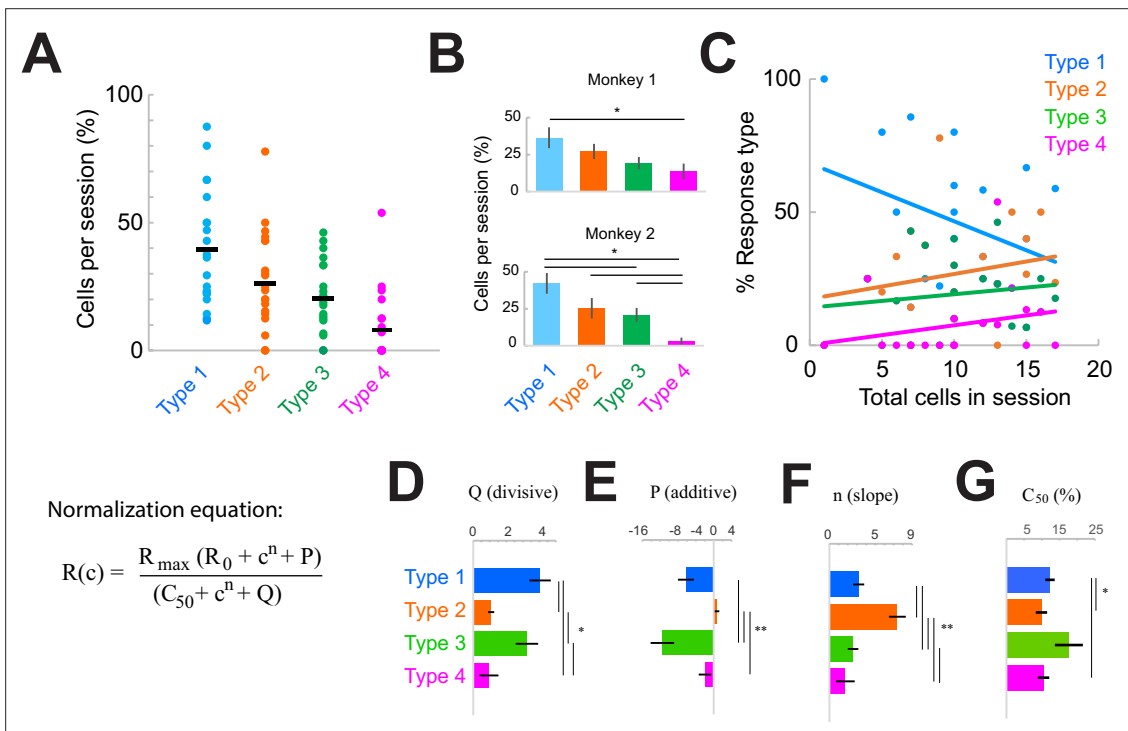

**Figure 3.** | Response type distributions are stable across sessions and subjects. (**A**) Proportion of neurons in each category type recorded in each session. Horizontal black line shows the mean across sessions. (**B**) Proportion of neurons in each category across sessions in each monkey. *p<0.05, Kruskal-Wallis test, post-hoc Tukey test. (**C**) Sample-size effect. Sessions with higher overall cell counts have a greater proportion of Type 2–4 cells and fewer Type 1 cells. The y-axis represents the percent of each response type as a function of the total number of cells recorded in each session (n = 22). Lines represent linear fits of each response type. (**D–F**) Normalization model fit parameters for each response type, showing the divisive (**D**) and additive (**E**) components as well as the slope (**F**). (**G**) Contrast that elicits 50 % of the maximum response (c50), measured from fits. Bars show the mean and standard error of each group. *p<0.05, **p<0.001, Kruskal-Wallis test, with post-hoc Tukey test.

The online version of this article includes the following figure supplement(s) for figure 3:

**Figure supplement 1.** Cross-session heterogeneity not explained by known experimental factors.

**Figure supplement 2.** Orientation preference and tuning sharpness are equally variant across Type 1 dominant sessions and mixed-type sessions.

all layers, while Type 2, 3, and 4 responses were primarily found in G and IG. These cumulative population results were confirmed by subsequent analyses restricted to sessions in which light-modulated units were simultaneously recorded across all three laminar divisions (n = 8 sessions, *Figure 2—figure supplement 3*). Further, given the considerable variability in the thickness of the granular layer across anatomical studies (ranging between 360 and 580 microns from our literature survey), to check whether our 400 -µm granular layer convention impacted the laminar distributions of response types, we also redefined granular as spanning 300 µm. However, this did not substantially alter our previous results (*Figure 2—figure supplement 3C, D*).

## Response heterogeneity can be explained by variations in network connectivity

Critically, all four response types were found simultaneously within individual recording sessions, when visual stimulus properties and light delivery parameters were identical (*Figure 3A*). This means that the overall diversity of responses cannot be explained by subtle differences in experimental parameters that can vary across sessions. All sessions with multiple simultaneously recorded neurons had more than one response type (data from individual sessions are shown in *Figure 3—figure supplement 1*).

The overall proportions of each response type were consistent within sessions (*Figure 3A*) and similar across monkeys (*Figure 3B*; p<0.0001, Kruskal-Wallis test, post-hoc Tukey test; within-type comparisons were not significantly different across monkeys, p>0.05). However, there was some variation in the types of responses observed day-to-day (*Figure 3A*, *Figure 3—figure supplement 1B*). Variability across sessions could be due to something trivial, such as sample-size effects (*Figure 3C*; rarer Type 3–4 responses were more likely to be found in sessions where more cells were simultaneously recorded), or due to other factors that could vary across sessions (*Figure 3—figure supplement 1*). To address these latter possible sources of heterogeneity across sessions, we considered whether the patterns of heterogeneity were correlated with four measurable factors. Sessions were organized either according to the proportion of Type 1 responses (*Figure 3—figure supplement 1*, left column) or in their chronological recording order (*Figure 3—figure supplement 1*, right column). We then asked whether the distribution of response types could be correlated with cumulative damage to the cortex (as indicated by the chronological order; *Figure 3—figure supplement 1F*), differences in laminar sampling (*Figure 3—figure supplement 1C, H*), recording depth (*Figure 3—figure supplement 1D, I*), or the identity of fiber optic used for light delivery on specific days (*Figure 3—figure supplement 1E, J*; laser power was the same across sessions). However, none of these factors correlated with heterogeneous response patterns observed in each session.

In primates, stimulus representation is anisotropically represented across the cortical surface. Our acute recordings sampling different neural populations could be located within different network orientation preference contexts of V1 (i.e., the column's proximity to pinwheel centers would influence the range of orientations represented in a given population). The visual stimulus orientation was optimized for the recorded cell population but could vary for individual cells within a session. We first examined whether the distribution of orientation preferences across neurons varied across sessions. More diversity of orientation preferences would be expected closer to pinwheel centers. However, we found no difference in either the distributions of orientation preferences observed in each session (*Figure 3—figure supplement 2A*; p = 0.497, t-test, two-way) or the orientation selectivity of the cells (indicative of the sharpness of tuning) across sessions (*Figure 3—figure supplement 2B*; p = 0.600, t-test, two-way).

Next, we examined the tuning of the response types (*Figure 3—figure supplement 2C, D*) but found no differences across types with respect to the difference between the cell's preferred orientation and that of the visual stimulus (*Figure 3—figure supplement 2C*; p = 0.06, Kruskal-Wallis test). Nor was there a difference across response types in the sharpness of tuning (*Figure 3—figure supplement 2D*; p = 0.12, Kruskal-Wallis test). Since cells within the suppressed column would also have largely overlapping orientation preferences, differences in orientation preference between the recorded and suppressed cortical columns fail to explain the heterogeneous responses found within sessions. Further, we did not measure any significant difference in population tuning properties across sessions, which would have been indicative of sampling functionally different portions of the V1 network. We conclude that differences in orientation preference do not adequately explain cross-session variability within our data.

Having ruled out the above factors, and given that suppressing the local network consistently produces the four basic simultaneous response types within a column, we reasoned that the response heterogeneity revealed here is most likely due to variations in functional connectivity patterns between the lateral network and individual cells within a distal cortical column. We examined this issue by quantifying the differences in the normalization model fit parameters associated with each response class (*Figure 3D–G*). Specifically, model parameters *P* (additive, *Figure 3E*) and *Q* (divisive, *Figure 3D*) represent the input provided by the local network, while *n* (slope, *Figure 3F*) and *c50* (semi-saturation constant; measured from the fits in *Figure 3G*; model parameter value in *Figure 2—figure supplement 2G*) represent the intrinsic stimulus responsivity. Note that *P* and *Q* are measured on the trials with optogenetic suppression ('laser trials'). This analysis revealed distinct stimulus response properties associated with each class. For example, Type 1 cells were sensitive to a broader range of stimulus contrasts (evident by the lower slope, *Figure 3F*, and higher c50, *Figure 3G*) and were more strongly modulated by the local network (*Figure 3D–E*) compared to Type 2 cells. Type 1 cells were also less sensitive to low-contrast stimuli compared to Type 2 or Type 4 cells (evident from the higher c50 value, *Figure 3G*). The Type 2 class displayed the lowest values for the network inputs (*Figure 3D–E*) compared to other cell classes. We also quantified the portion of cells in each type that did not exhibit saturating responses at high contrasts (*Figure 2—figure supplement 2E, L*), and thus maintained sensitivity in that range. These cells comprised the majority of Type 3 responses (54.8%) and were least represented in the Type 2 group (12.2%) (*Figure 2—figure supplement 2H, I*). Interestingly, the stimulus responses (*Figure 3F–G*) of Type 1, 3, and 4 classes were statistically similar but varied mostly in their additive and divisive network inputs (*Figure 3D–E*).

Altogether, these results suggest that intracortical connections control the heterogeneity of responses within the local network. This is surprising for two reasons. First, they reveal that normalization by the local population is not uniformly applied on columnar neurons. That is, individual neurons receive different degrees of normalization from their local network. Second, the presence of four consistent response patterns (Types 1–4) suggests that normalization could come in four 'motifs' mediated by four underlying connectivity profiles between columnar neurons and their local network. We further asked what type of intracortical interactions could possibly explain the four response types observed after the optogenetic suppression of neurons located at distal locations.

## Heterogeneous response types can be captured by a simple network model

We devised a small-scale firing-rate model in which the response of a model neuron is determined by a linear combination of feedforward excitatory drive and local recurrent excitatory and inhibitory inputs (*Figure 4A*). The output neuron represents the experimental recording site at the distal location (*Figure 1D*, left), while the local network represents the activity of the suppressed network at the light source (*Figure 1D*, right). The optogenetic suppression of the nearby network was modeled as a constant reduction in the firing rates of network excitatory cells (equivalent to a 12.9 sp/s decrease in firing, as observed experimentally; *Figure 1E*). We tested the impact of two possible factors that could differentially modulate neuronal activity to generate the heterogeneous response types observed experimentally. First, the stimulus sensitivity of the network drive to individual cells could vary, i.e., the different response types could be due to differences in the cumulative contrast response function of the normalization pool impacting each neuron. Here, the sensitivity variable, implemented as a multiplicative gain, refers to the dynamic range of the stimulus response across contrasts (*Figure 4A*, right). Second, the balance of excitatory and inhibitory currents ('E/I ratio') could vary slightly across response types. In our model, the amount of inhibitory synaptic current was determined by scaling the excitatory current by the E/I ratio. Since E/I ratio is tightly controlled in cortical networks, we hypothesized that even minor fluctuations could produce the diversity of observed responses.

Our crucial observation was that modulating local network sensitivity alone (while keeping feedforward input and E/I ratio fixed) can reproduce all four types of responses observed experimentally (*Figure 4B–E*). In other words, for a given feedforward input (such that from the lateral geniculate nucleus), our model neuron could be coaxed to reproduce all four response motifs when the local network input was suppressed. Which response motif was modeled depended on how strongly the network input was driven by the stimulus. We also tested the effect of altering the sensitivity of the feedforward gain (rather than the local excitatory network gain), but this manipulation did not

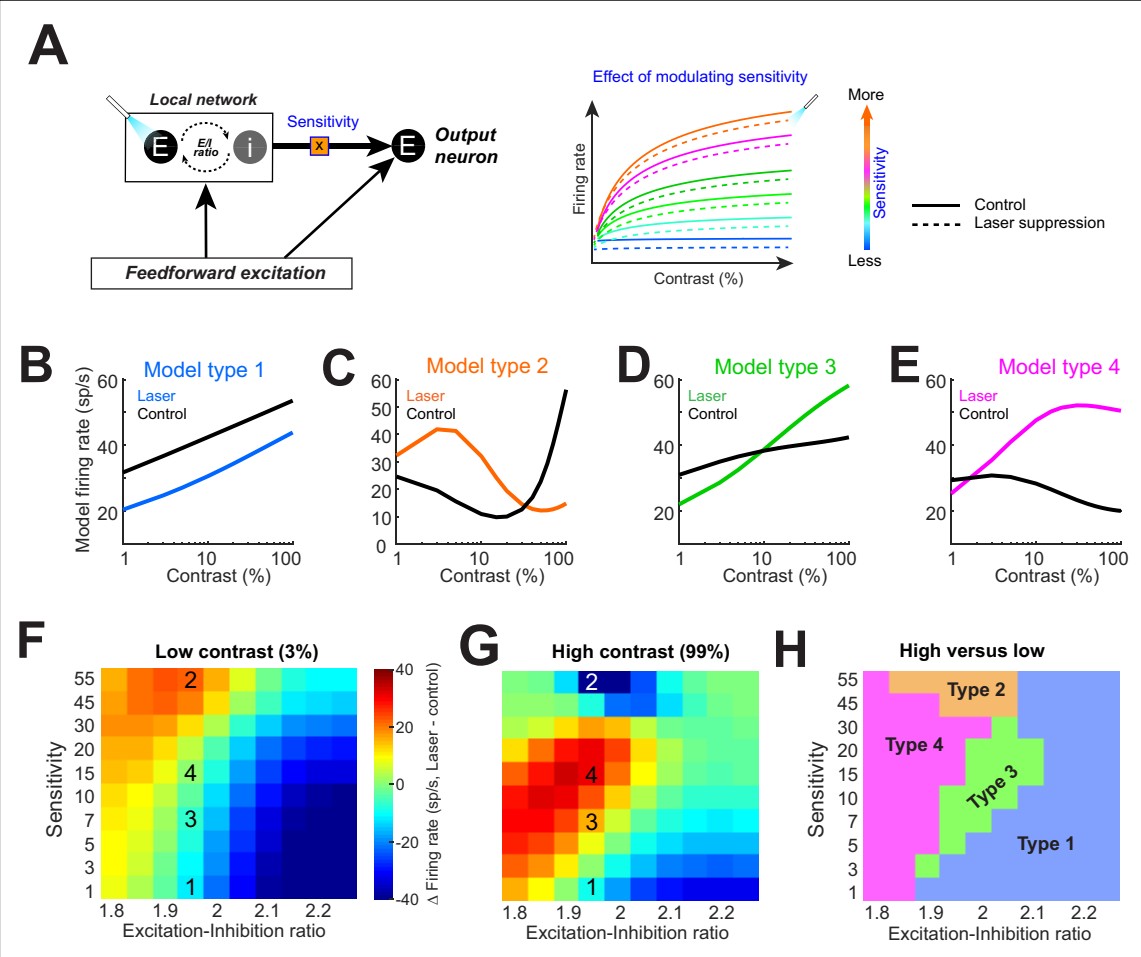

**Figure 4.** Model replicates heterogeneous responses to optical suppression of local network by altering network stimulus sensitivity. (**A**) Cartoon model (left). The response of the output neuron is calculated as a linear summation of a feedforward and a local network component. Feedforward drive is purely excitatory, while the local network is represented as a mixture of excitation and inhibition currents. Inhibition followed excitation according to a fixed ratio ('E–I ratio'). The gain of the excitatory current is modulated by a multiplicative parameter ('sensitivity'). (Right) The firing rate of the output neuron as network sensitivity is varied. Solid colored traces show model responses on control trials. Dashed lines represent optogenetic suppression trials. (**B–E**) Varying the sensitivity of the network input while holding E/I ratio stable produces the four basic response motifs observed experimentally. The parameters used to generate these responses are shown in panels (**F, G**). (**F, G**) Differences in the model firing rate between laser and control trials for low-contrast (**F**) and high-contrast stimuli (**G**), while varying the local network sensitivity (y-axis) and E/I ratio (x-axis). Overlaid numbers represent the parameter combinations for generating the response types in panels (**B–E**). (**H**) Boundaries of parameter space dividing the four response types based on changes in firing rate associated with optogenetic suppression for low (**H**)- and high (**G**)-contrast conditions.

The online version of this article includes the following figure supplement(s) for figure 4:

**Figure supplement 1.** Modulating sensitivity of model feedforward input is insufficient to reproduce experimentally observed effects.

reproduce all four experimentally observed response motifs (*Figure 4—figure supplement 1*). Next, we examined the combined effects of altering both the network sensitivity and E/I ratio. *Figure 4F–G* shows the difference in the model neuron responses between control and laser-suppression conditions across the range of sensitivity and E/I ratio parameters tested, both for the low (*Figure 4F*)- and high (*Figure 4G*)-contrast stimuli (the superimposed numbers on the heatmaps correspond to the conditions under which the examples in *Figure 4B–E* were obtained; '1' refers to model Type 1 in *Figure 4B*, etc.). The difference in responses between the low (*Figure 4F*)- and high (*Figure 4G*)-contrast conditions yielded a feature-space map revealing the combinations of sensitivity and E/I ratios under which various response types can be generated (*Figure 4H*). We found that for a limited range of E/I ratios (1.95–2), all four response types can be generated solely by altering the local network sensitivity. However, there is no equivalent range of network sensitivity values that could produce all four response types if only the E/I ratio is modulated.

These modeling results indicate that differences in the stimulus sensitivity of network inputs to individual neurons, located within a column, are sufficient to account for the heterogeneous contrast responses experimentally observed following inactivation of the lateral network (*Figure 1F–I*), but differences in E/I ratios are not. When the network sensitivity is low, Type 1 responses dominate, but as stimulus sensitivity increases, Type 2, 3, and 4 responses emerge. Since Type 1 responses are dominant, the majority of neurons receive network input with relatively low stimulus sensitivity. This is consistent with the standard normalization model, where the normalizing pool consists of a broadly tuned population. Surprisingly, our results demonstrate that the normalization is applied idiosyncratically to neurons within a population and can account for the heterogeneous continuum of experimentally observed responses. One limitation of our simple model is that it does not distinguish between individual network inputs but represents the cumulative synaptic current from the entire local network. Further, our experiments only utilized static oriented gratings of varying contrasts. Thus, it is quite possible that varying other visual features represented in V1 (such as spatial frequency or color) could reveal different patterns of local connectivity.

## Relationship between response heterogeneity and behavioral performance

We next asked whether the heterogeneity of neural responses observed during optogenetic suppression of the distal network can influence behavior. The prevalent idea in neuroscience studies is that inactivating sensory cortical responses should yield a marked change, typically suppressive, in behavioral performance. However, our results in *Figures 1–3* raise the issue of whether the diversity of response changes during cortical inactivation could be associated with diverse changes in behavioral performance. We directly examined this issue by training monkeys to perform a contrast detection task (see 'Materials and methods') while inactivating cortical responses as previously described (*Figure 1C*). Specifically, animals detected oriented gratings presented for 300 ms at various contrasts. While the orientation of the stimulus was optimized for the distal recording site, the size of the stimulus was large enough to cover the receptive fields of both the direct and indirect sites, and thus the activity at both sites could contribute to the behavioral choice. Control and laser trials, and contrast conditions were randomly interleaved. Laser stimulation was delivered as one continuous pulse for 300 ms and was synchronized with the visual stimulus.

Across daily sessions and contrast conditions, we found that behavioral performance was significantly changed on 57.9 % of laser suppression trials compared to control trials (31.6 % impaired and 26.3 % facilitated; *Figure 5—figure supplement 1A, B*). This raises the possibility that these diverse changes could be explained, at least in part, by the diversity of neural responses outside the target inactivated area. To investigate how the changes in behavioral performance correlate with the observed neural activity changes following optogenetic inactivation, we divided sessions based on the proportions of response types within individual sessions, and if more than 50 % of the recorded units in a session were categorized as Type 1, we labeled the session as Type 1 session.

First, we examined the sessions containing predominantly Type 1, purely suppressive, responses at distal cortical locations (*Figure 5A*; p = 2.30 × 10⁻⁶, one-way Kruskal-Wallis test, df = 3, Chi-sq = 28.94, post-hoc t-test). Across the 10 sessions dominated by Type 1 responses, we found that detection performance was impaired on laser trials (target reports on laser trials: 9.3 ± 1.9%; control trials: 19.5 ± 5.1%, mean ± sem) for the lowest contrast stimuli (*Figure 5B*; p = 0.04, t-test, paired, two-way), while detection performance did not change during the blank condition (*Figure 5C*) or for higher contrasts (p>0.1, *Figure 5—figure supplement 1C*). The fact that behavioral changes are only observed for the low-contrast stimuli is not surprising and is consistent with previous studies (*Afraz et al., 2006*; *Andrei et al., 2019*; *Bisley et al., 2001*; *Ditterich et al., 2003*), as sub-optimal stimuli better reveal subtle changes to psychometric curves. Second, we examined the remaining sessions (n = 11) with more balanced proportions of the four response types, with Type 1 responses occurring as often as other response types (*Figure 5D*; Types 1–3 are uniformly distributed, with more Type 2 than Type 4 cells, p = 0.022, one-way Kruskal-Wallis test, df = 3, Chi-sq = 9.67, post-hoc t-test). However, in the sessions in which V1 responses were highly heterogeneous, detection performance remained unchanged between laser and control trials both for the low (*Figure 5E–F*; p = 0.40, t-test, paired, two-way) and high contrasts (*Figure 5—figure supplement 1D*). Importantly, performance on control trials was not significantly different between Type 1 dominant sessions and the remaining sessions

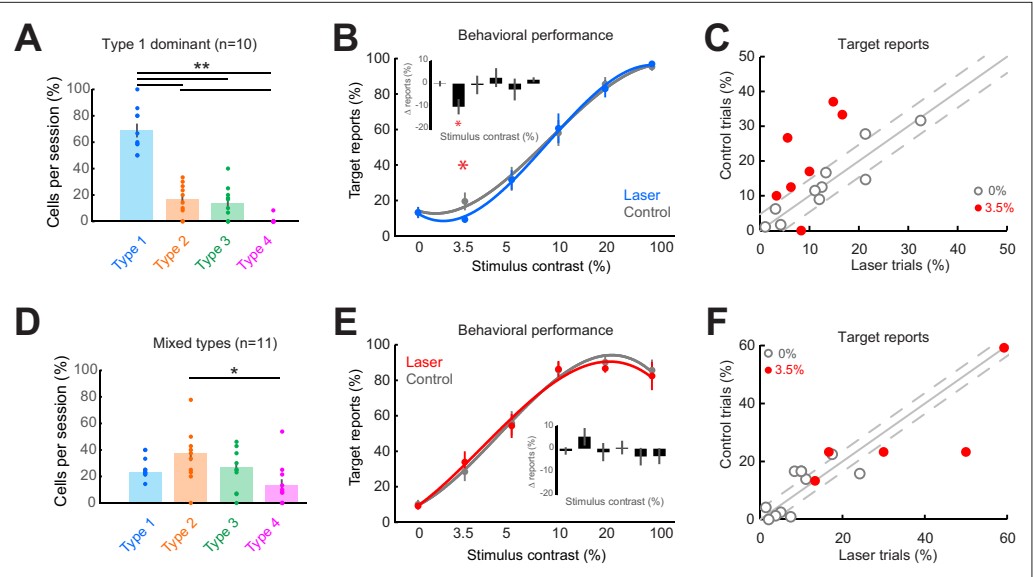

**Figure 5.** Consistent impairment in behavioral performance occurs only when optogenetic inactivation is associated with homogeneous response suppression in V1. (**A**) Proportions of cell response types found in those sessions containing >50 % Type 1 responses. Dots show individual sessions. Bar graphs show mean ± sem. **p = 2.30 × 10⁻⁶, one-way Kruskal-Wallis test (df = 3, Chi-sq = 28.94, post-hoc t-test). (**B**) Task performance in Type 1 dominant sessions showing % trials in which animals reported the presence of a visual stimulus on laser (blue) and control (gray) trials. *p = 0.04, t-test, paired, two-way. Session counts per contrast, ascending (left to right), are 10, 7, 10, 9, 3, and 8. Points show mean ± sem. Fits are third-order polynomials. Inset bar graph shows performance difference (laser minus control) across contrasts. (**C**) Target reports for Type 1 dominant sessions (n = 10) for 0 % (open gray circles) and 3.5 % (red filled circles) contrast. The unity line (solid gray diagonal) shows ± sem behavior responses across all contrasts in control condition (flanking dashed diagonal lines). (**D**) Same conventions as in panel (**A**), but for the sessions with mixed response types; *p = 0.022, one-way Kruskal-Wallis test (df = 3, Chi-sq = 9.67, post-hoc t-test). (**E**) Same conventions as in panel (**B**), showing no change in behavioral performance across laser (red) and control (gray) trials. Session counts per contrast, ascending (left to right), are 11, 5, 10, 11, 6, and 9. (**F**) Same conventions as in panel (**C**), but for mixed-type dominant sessions (n = 11).

The online version of this article includes the following figure supplement(s) for figure 5:

**Figure supplement 1.** Behavioral task performance was heterogeneously affected on optogenetic suppression trials.

**Figure supplement 2.** Eye position and pupil size do not account for differences in behavioral performance between Type 1 and mixed-type sessions.

---

at this contrast (p = 0.33, two-way t-test). For a subset of these mixed-type sessions, where Type 2 responses were dominant ( >35% of Type 2 cells recorded, n = 6 sessions; *Figure 5—figure supplement 1F*, p = 0.0032, one-way Kruskal-Wallis, df = 3, Chi-sq = 13.8, post-hoc t-test), we found a significant impairment in detection performance at the 20 % contrast in laser trials compared to control (*Figure 5—figure supplement 1E, G, H*; p = 0.016, t-test, paired, two-way), consistent with the range of contrasts for which the Type 2 cells are suppressed by light (*Figure 2A and B*). Furthermore, although strongly facilitated detection performance was often evident (*Figure 5—figure supplement 1A*, points below the diagonal), facilitated responses were not systematically consistent across sessions and contrasts, and did not meet our statistical criteria for significance. We could not assess the behavioral performance for Type 3 or Type 4 response types because those types were rare and were always outnumbered by Type 1 or Type 2 responses.

As small changes in eye position could have a significant impact on V1 responses (*McFarland et al., 2016*), we examined whether the behavioral performance differences between the two types of sessions that we analyzed separately (*Figure 5B and E*) could be accounted for by significant differences in eye position (*Figure 5—figure supplement 2*). However, across sessions and stimulus conditions, we did not observe a significant difference in eye position (*Figure 5—figure supplement 2A, B, D, E*; median p-value = 0.493, Wilcoxon ranked sum test) or pupil size (*Figure 5—figure supplement*

*2C, F*; median p-value = 0.498, Wilcoxon ranked sum test) between laser and control conditions. Importantly, there were no statistically significant differences (p>0.1) in eye movements and pupil size between Type 1 dominant and mixed-type sessions. Rather, both fixation locations (see 'Materials and methods'), eye movements, and pupil size were remarkably stable across sessions. Altogether, these results indicate that consistent behavioral changes were only observed when neuronal responses at the optogenetically targeted site and at the off-target distal sites were both suppressed. When the distal neural population is dominated by heterogeneous changes in firing rates, changes in behavioral choices are mixed and inconsistent across sessions.

## Discussion

Contrary to the commonly held assumption that cortical inactivation has a negligible impact on surrounding cortical tissue, our study reveals highly heterogeneous changes in neural population responses hundreds of microns away. The most surprising effect is the uncovering of response facilitation and mixture of facilitation and suppression in the middle and deep cortical layers of the distal network, with significant consequences for behavioral performance. Since the laminar structure is ubiquitous across the sensory cortex, these results may be broadly applicable across multiple cortical areas and species.

Our study demonstrates that focal optogenetic suppression induces complex changes in the functional properties of nearby neural populations. Contrary to our expectation, cortical inactivation did not result in a simple distance-based decay effect of suppression as a function of proximity to light source. Instead, we found a mix of facilitatory and suppressive effects at distal cortical sites surrounding the directly inactivated region, which were related to changes in behavioral performance. Indeed, the relationship between potential off-target effects of cortical inactivation and behavior has not been previously studied. Previous causal investigations assumed that cortical inactivation silences neurons in the target area while the surrounding tissue is only negligibly impacted. However, our findings that off-target side effects are highly heterogeneous complicate the interpretations of inactivation manipulations, especially when they are related to changes in behavior.

We have shown that behavioral performance is consistently impaired only when the net activity of both on- and off-target sites is suppressive (*Figure 5B*). In contrast, heterogeneous off-target responses result in heterogeneous changes in behavioral performance (*Figure 5—figure supplement 1G*). Furthermore, complex changes in responses during suppression of distal cortical activity were predominantly present in deeper cortical layers, and hence studies limiting electrophysiological (*Nassi et al., 2015*; *Sato et al., 2014*) and optical recordings (*Michel et al., 2018*) to superficial layers would underestimate these important network effects. We suggest that the mixed off-target effects found here could possibly explain the highly variable changes in behavioral performance during optogenetic inactivation, particularly observed in non-human primate studies (*Dai et al., 2014*). Therefore, new strategies need to be developed to limit, or at least monitor, such off-target network effects. One potential approach for future inactivation studies, particularly those involving behavioral measurements, would be the use of a secondary recording electrode distal to the targeted illumination site to record any side effects produced in the local network.

Furthermore, our results provide strong experimental support for prior untested hypotheses that the local network is a source of normalizing drive to cortical neurons. Normalization has been proposed to be one of the canonical computations in the cortex (*Carandini and Heeger, 2011*); yet it presently lacks a clear mechanism. We have shown here that the local network does indeed provide normalizing inputs to neurons within a column. However, this normalizing input is not uniform, but rather it is idiosyncratically tailored for each neuron. We have shown that the heterogeneous normalizing effects fall along a continuum, with four basic motifs. This diversity of normalizing responses has not been previously addressed. Type 3 and Type 4 response motifs, in particular, with their rather surprising instances of response augmentation at high contrasts, have never been previously described. These four types of responses might serve functionally distinct roles in stimulus encoding. For example, normalization has been shown to stabilize neural responses and thus reduce variability to repeated stimulus presentations (*Coen-Cagli and Solomon, 2019*). The heterogeneous responses may also reflect the mechanisms by which the network retains sensitivity over the entire range of contrasts, despite the fact that most cells saturate their responses at high contrasts. In other words, although our model is able to generate all four responses solely by modulating the local network inputs to a single cell, it is also

possible that the heterogeneity could arise from differences in the intrinsic contrast response properties across cells. Further work is required to understand why the local network modulates neurons within a column so differently.

Heterogeneity of normalization has not been well studied, nor is it well defined for populations of neurons. The denominator of the normalization equation represents the cumulative activity of the local network (or 'normalization pool'). Since the model is implemented for one neuron at a time, the model is agnostic to whether the normalization pool is the same or different for individual neurons. At the site of light application, *Nassi et al., 2015* showed heterogeneous responses, owing to evoked normalization in the network, but because neurons were mostly recorded one at a time, they could comment on the source of the heterogeneity, since all responses were not simultaneously present.

Our experimental results are focused on columnar interactions separated by a short range of about 300 µm distance. However, it is quite possible that some of the heterogeneity is produced by network connections that span much further. Computational studies have found that tuning curves can be sharpened when short-range connections are primarily excitatory and longer-range connections are predominantly inhibitory, creating Mexican-hat-shaped interactions between columns (*Ben-Yishai et al., 1995*; *Spiridon and Gerstner, 2009*). It is quite possible that some of the effects recorded at 300 µm distance could be due to activity changes that occurred at a longer distance ( > 500 µm) primarily mediated by long-range inhibitory projections.

The degree of off-target response heterogeneity exhibits session-by-session variability. Indeed, although the types of neuronal responses outside the directly inactivated cortical region were consistent across sessions and monkeys (*Figure 3A–B*), there was variability in the distribution of response types, that is, some sessions were dominated by Type 1 responses, whereas others contained more mixed response types (*Figure 3—figure supplement 1*). This variability could be partially attributed to random sampling inherent in acute electrophysiological recordings (*Figure 3C*; see also *Figure 3— figure supplement 1*). However, the fact that we observed a significant difference in task performance between the Type 1 dominant sessions relative to the more heterogeneous 'mixed-type' sessions strongly suggests that these cross-session variations are due to differences in local circuit connectivity patterns (*Chelaru and Dragoi, 2008*; *Keller and Martin, 2015*). Indeed, it has been shown that the responses of nearby neurons located within the same functional column exhibit a high degree of heterogeneity in their receptive field properties (*Hubel and Wiesel, 1962*; *Lee et al., 1998*; *Ringach et al., 2002*), and this heterogeneity has been frequently linked to the heterogeneity of local excitatory and inhibitory intracortical circuits (*Chelaru and Dragoi, 2008*; *Landau et al., 2016*; *Padmanabhan and Urban, 2010*). Additionally, while our model (*Figure 4*) shows that for a given E/I ratio, all four response types can be generated, excitatory and inhibitory currents can be differentially tuned, and the E/I ratio can vary depending on stimulus orientation (*Wilson et al., 2018*). In our model, changing the E/I ratio alters the distribution of response types observed (*Figure 4H*), which could also explain the variations of response types across sessions. This raises the possibility that heterogeneities in the strength of local intracortical inputs to V1 neurons, which control the different response types after optogenetic suppression of the distal network, can explain the 'cross-session' variability in response types shown here.

The exact network mechanism involved in generating the response heterogeneity revealed here is unknown. We reasoned that the source of contrast gain modulation at off-target sites after optogenetic inactivation is the change in the strength of local network inputs. In mouse visual cortex, parvalbumin-positive (PV) interneurons are believed to be critically involved in the gain modulation of nearby pyramidal neurons (*Atallah et al., 2012*; *Wilson et al., 2012*). However, PV interneurons in mouse auditory cortex do not modulate the overall gain of local pyramidal cells in a stimulus-dependent manner, as would be required by contrast normalization mechanisms (*Cooke et al., 2020*). The basis of our heterogeneous responses in the lateral network could be explained by the horizontal connections made by PV interneurons. For instance, axons of basket cells (morphologically defined PV cells [*Scala et al., 2020*]) in cat visual cortex can target neurons up to 1360 µm away and do not obey orientation domain boundaries, with 57 % of projections synapsing at sites with orientation preferences differing by >30° from the location of the basket cell's soma (*Kisvárday et al., 1994*). One possibility is that the optogenetic suppression of excitatory neurons in our experiments might have led to a decreased drive to nearby PV cells, which in turn reduces inhibition on both local and distant pyramidal cells that they target. The heterogeneous responses

in the lateral network could thus result from a combination of local excitatory and inhibitory inputs. This is captured by our model (*Figure 4*), which shows that varying the net response gain ('sensitivity') of the local network input to a model cell can reproduce all four response types. This hypothesis is consistent with the observed continuum of responses observed within each response class (*Figure 2B*).

Previous studies silencing cortical populations, particularly those using optogenetic tools, have primarily focused on the light-targeted zone without investigating collateral effects on the nearby local network (*Diester et al., 2011*; *Fetsch et al., 2018*; *Han et al., 2011*; *Nassi et al., 2015*; *Ohayon et al., 2013*; *Trouche et al., 2016*). *Crook and Eysel, 1992* have previously shown that inactivating visual cortex, using gamma-aminobutyric acid (GABA) microiontophoresis, could substantially broaden tuning curves of neurons located 600 microns away, but only if the inactivated site had a preferred orientation drastically different from the recording site. A recent study by *Li et al., 2019* examined the effects of cortical inactivation on more distant cortical sites by comparing the spatiotemporal profiles of several suppressive opsin constructs in awake, but otherwise passive, mice. This study thus measured differences in spontaneous activity in the absence of sensory inputs. Similarly, we found that in the absence of a visual stimulus (the 0 % contrast condition), firing rates in the distal network are not significantly influenced by optogenetic suppression (*Figure 1F–I*, *Figure 2—figure supplement 1*). Heterogeneous responses emerged only when optogenetic suppression was paired with a visual stimulus. This detail is important because the presence of stimulus in visual cortex activates otherwise minimal local network connections to produce characteristic response features such as contrast normalization and near-surround suppression. Further, stimulus-paired optogenetic manipulations are commonly employed in experimental designs that closely monitor behavioral responses (*Acker et al., 2016*; *Andrei et al., 2019*). Our findings demonstrate that focal optogenetic suppression has unpredictable effects in the surrounding network. While this finding provides a potent method with which to investigate local circuits, it adds complexity to interpreting the causal relationship to behavior.

## Materials and methods
### Animal subjects
This study was performed in strict accordance with the recommendations in the Guide for the Care and Use of Laboratory Animals of the National Institutes of Health. All of the animals were handled according to the approved institutional animal care and use committee (IACUC) protocols of the University of Texas, Houston. The protocol was approved by the Committee on the Ethics of Animal Experiments of the University of Houston (protocol number: AWC-20–0075). Two male rhesus monkeys (*Macaca mulatta*) were trained for a contrast detection task and were implanted with a 19 mm titanium chamber (Crist Instruments) over V1. All surgery was performed under general anesthesia, and every effort was made to minimize suffering.

### Viral vector preparation and injection
The anion channelrhodopsin *Gt*ACR2 (*Forli et al., 2018*; *Govorunova et al., 2015*; *Mauss et al., 2017*; *Mohammad et al., 2017*) was packaged in a lentiviral construct under the control of the promoter α-CamKII (LV-CaMKIIα-GtACR2-GFP). The vector was prepared by the University of North Carolina Vector Core, with a titer of $10^9$ IU/ml. Four V1 columns, and five sites per column (spaced every 250 μm), were targeted for injection per animal. The virus was injected using a 29 gauge needle attached to a Hamilton syringe. The syringe was advanced using a computer-interfaced micromanipulator (NAN Instruments). For each injected column, the needle and syringe were lowered to the deepest site at which neural activity had been recorded in the previous days. After a 15 min waiting period (5 min waiting period for every subsequent site within the column), the plunger was independently advanced, delivering 1 μl of the viral suspension over a 10 min period. After a 5 min waiting period, the needle was retracted upward to the next site. The process was repeated for each additional site within the column and then for each additional column. A custom-built grid (Crist Instruments) was used during the injection as well as during subsequent recordings to precisely target the tissue expressing the virus.

## Cortical biopsy and immunohistochemical tissue staining

In order to avoid unnecessary animal sacrifice, we developed a method to biopsy cortical tissue in awake macaques. After concluding all experiments, a transfected column of the cortex was biopsied using an aseptic aspiration technique. A small amount of lidocaine was applied to the cortical surface, and an 18 gauge needle attached to a syringe filled with phosphate-buffered saline (PBS) was positioned over the biopsy site (recording site significantly modulated by optical stimulation). The syringe was then lowered to a predetermined depth (previous recording site with laser-responsive cells) with a computer-interfaced micromanipulator (NAN instruments). The syringe was then rotated in order to cut surrounding tissue connections. After a 5 min waiting period, the plunger was manually retracted at a very slow speed until tissue is apparent in the PBS of the syringe. The syringe and needle were immediately retracted and the biopsied tissue was fixed in paraformaldehyde (1 % solution) for 6 hr. This technique provides an adequate amount of tissue with minimal impact to the animal.

Tissue was fixed in paraformaldehyde (4%) in PBS for 1 hr and then cryoprotected in sucrose (30%) in PBS at 4 °C overnight before sections were cut (20 µm). Sections were incubated with primary antibodies against NeuN (polyclonal Guinea pig [Synaptic Systems, Göttingen, Germany], used at 1:250), CamKIIα (monoclonal mouse, used at 1:500 [Dr. Neal Waxham (UT Houston)]), and GFP (polyclonal rabbit [Synaptic Systems, Göttingen, Germany], used at 1:1000) at 4 °C overnight. Sections were incubated with secondary antibodies against the primary antibodies for NeuN (Alexa 647-conjugated goat anti-Guinea pig IgG, used at 1:300), CamKIIα (Alexa 568-conjugated goat anti-mouse IgG, used at 1:300), and GFP (Alexa 488-conjugated goat anti-rabbit IgG, used at 1:300) at room temperature for 1 hr. All secondary antibodies were obtained from Thermo Fisher Scientific. Sections were mounted with Vectashield mounting medium with 4', 6-diamidino-2-phenylindole (DAPI) (Vector Labs). Images were obtained using a Zeiss LSM800 confocal microscope and analyzed using the ZEN 3.0 software.

## Behavioral experiments

Monkeys completed a contrast detection task using sine-wave gratings presented at recorded cell population's preferred orientation and a fixed spatial frequency. Stimuli were generated using Matlab (Mathworks) with Psychophysics Toolbox (*Brainard, 1997*) and were presented on a gray screen 19 inch cathode ray tube(CRT) monitor, located 90 cm away from the animal. On each trial, animals held a response lever and fixated on a central point (0.5°, with 1° fixation window). If fixation was broken before the end of the trial, or if the response lever was lifted before the response period, the trial was aborted. Eye position was continuously monitored using an infrared eye tracking system (EyeLink II, SR Research) with a 1 kHz sampling rate. On 50 % of trials, a stimulus was presented over the receptive fields of recorded cells. Stimuli were presented at four contrast levels per session (five total contrasts were used across sessions: 3.5, 5, 10, 20, and 100%) on a gray background. Laser onset was synchronized with the stimulus onset, which both lasted 300 ms (light was continuously on during this period). Stimuli had a median size of 2.5° (± 0.6°) and a spatial frequency of 2 cycles per degree. Additionally, on half of the trials, optical stimulation was delivered simultaneously with the visual stimulus (or when the visual stimulus would have appeared on a no-stimulus trial) and was delivered equally throughout all stimulus contrast conditions. During the response period (600–1800 ms after stimulus onset), animals released the bar to signal the presence of a stimulus or continued to hold the bar to signal the absence of a stimulus in that trial. Correct behavioral responses were rewarded with five juice drops. Animals completed 128–480 trials per session (inter-trial intervals were dependent on the animal and averaged 5.0 s ± 4.7 sem).

## Electrophysiology

Extracellular recordings were completed using 16 or 24 channel laminar electrodes (U-probe, Plexon Inc), advanced by a computer-interfaced micromanipulator (NAN Instruments). Contacts were 25 µm in diameter, equally spaced (100 µm intercontact spacing) and coated in platinum iridium. Neural activity was recorded in real-time using a Multichannel Acquisition Processor system (MAP, Plexon Inc). For 20 of the 21 sessions, one laminar electrode was inserted into V1 and a separate optical fiber (see below) was inserted approximately 300 µm away from the electrode. In this manner, the optical fiber silenced one population of cells, with no direct effects on the population recorded by the laminar electrode. For 1 of the 21 sessions, two laminar electrodes were inserted, such that one recorded the activity of cells being directly suppressed by the laser (probe was as close to the optical fiber as

possible) and of cells being indirectly suppressed by the laser (probe was about 300 μm away from the optical fiber). An additional seven sessions were recorded with the laminar electrode and the optical fiber as close as possible (~150 μm) in order to record the direct, light-induced suppression of *Gt*ACR2 activation. At the directly suppressed site, light-responsive cells were vertically clustered within 457.1 ± 104.3 μm (mean ± sem, n = 31 cells with maximum suppression within 30 ms of light onset), as measured by the maximum distance between electrode contacts for these units. This measurement includes both directly suppressed cells, and the nearby directly connected cells, as detecting suppression with millisecond precision is more difficult than detecting activation.

## Optogenetic stimulation

Optical activation of opsins was delivered via a 100 mW blue laser (473 nm; Laserglow Technologies) coupled to a penetrating optical fiber. The 200 -μm diameter optical fiber was encased in a stainless steel cannula for mechanical support and advanced into the brain using a computer-interfaced micro-manipulator (NAN Instruments). Prior to each experiment, the optical fiber was aligned to the upper third of recording contacts on the laminar electrode. Registration lines drawn onto the shafts of the electrode and fiber optic ensured that this spacing was preserved during recordings. Laser stimulation parameters were controlled by a waveform generator (Agilent Technologies). Optical stimulation was delivered as one continuous pulse for 300 ms and was synchronized with the visual stimulus. Light power output at the end of the fiber optic was periodically measured using an integrating sphere sensor (S124C, Thor Labs) and was kept consistent across sessions (~7–14 mW per mm$^2$). Importantly, the optical fiber was scissor-cut and the direction of the light was perpendicular to the cortical surface, meaning that the maximum light power was vertical, parallel to cortical columns. The lateral position of the fiber optic was approximately 300 μm away from the probe, such that the optical stimulation affected a nearby but distinct population of cells relative to the recorded responses. The spacing between devices was achieved using guidetubes placed inside a custom grid that sat inside the chamber.

## Cell classification

Units were sorted offline (Plexon Offline Sorter). On average, per session, we were able to identify 5 (± 4) single units and 7 (± 3.5) units (single or multiunit activity) whose activity could be significantly modulated by light application. Cells that were significantly modulated by light were identified by comparing activity in trials with optical stimulation and trials without optical stimulation. The transient response (0–150 ms of optical stimulation) and sustained response (150–300 ms of optical stimulation) were independently analyzed. If either the transient or the sustained response during optical stimulation was significantly different (p<0.005, Wilcoxon ranked sum test, with Bonferroni correction for multiple comparisons) from the control activity during the same time period, it was determined that the cell could be modulated by optical stimulation and was therefore considered light responsive.

## Layer identification

CSD was performed to identify cortical layers of V1 (*Hansen et al., 2012*; *Nigam et al., 2019*; *Schroeder et al., 1998*). Evoked response potentials in response to a high-contrast stimulus were recorded from equally spaced laminar contacts. The second spatial derivative of the evoked response potentials was computed (iCSD toolbox [*Pettersen et al., 2006*] for Matlab) and the granular layer was identified by finding the first sink, measured in nA/mm$^3$. Channels located in the primary sink were identified as the granular layer, while channels above the sink were identified as the supragranular layer and channels below the sink were identified as the infragranular layer.

## Contrast response functions and normalization fits

For all light-responsive units, we considered the 150 ms epoch containing the maximum stimulus response. To identify this epoch, stimulus responses on control (no laser) trials were divided into four non-overlapping 150 ms bins, aligned with the stimulus onset, and the bin with the maximum spike count across trials was identified. We then compared the responses on laser trials during the same epoch. For the zero-contrast (no-stimulus) condition, the epoch that most often yielded the strongest response during stimulus present trials was used. This variable epoch approach was preferred over a fixed epoch, which assumes neural responses to occur within a prescribed interval. Due to the wide

range of visual contrasts utilized, many units displayed clear shifts in response latency as a function of stimulus contrast (visible in **Figure 1F–I**), a well-known phenomenon (**Carandini et al., 1997**). This approach allowed us to tailor our analyses to the peak response of each neuron. For the majority of units (72.9%), the peak response averaged across stimuli occurred in 150–300 ms after stimulus onset.

We then fit the contrast responses with the normalization equation, using a procedure similar to that by **Nassi et al., 2015** and **Sato et al., 2014**. We first fit the firing rates in the control (no laser) condition with a hyperbolic ratio function using a least-squares method,

$$r(c) = r_0 + r_{max}\frac{c^n}{c_{50}^n + c^n},\tag{2}$$

where r0 is the baseline firing rate, $r_{max}$ is the maximum firing rate, c50 is the semi-saturation constant, and n is the slope of the function. Then, for laser trials, we used the same c50 and n parameters and solved for the remaining parameters, *P* and *Q*, using the full normalization equation (**Equation 1**). Quality of fits was assessed as the percentage explained variance, where $R^2 = [1-(\text{error sum of squares/total sum of squares})]$.

## Orientation selectivity

Orientation tuning for the recorded population was measured separately prior to the contrast detection task with optogenetic suppression. For this task, monkeys fixated on a central spot on the computer screen, while a movie stimulus consisting of a sequence of 48 circular 100 % contrast sinusoidal gratings (eight equidistant orientations randomly flashed at 30 Hz) was presented for a total duration of 1.6 s. The size and location of the stimuli were kept identical to the ones used in the subsequent detection task. The preferred orientation and orientation selectivity index (OSI) for each neuron were computed from Fourier components extracted from the orientation tuning curves as described previously (**Dragoi et al., 2002**). Variance of tuning across the population of units in individual sessions was computed using circular statistics (CircStat toolbox for MATLAB [**Berens, 2009**]).

## Computational model

We built a firing-rate model of a single output neuron receiving excitatory and inhibitory synaptic currents from the local network and driven by a feedforward excitatory input. The total synaptic current, *Is*, is given by

$$\tau\frac{dI_s}{dt} = -I_s + aR_{ex} - bR_{ix} + I,\tag{3}$$

where $\tau$ is a time constant, *a* and *b* are positive coefficients, $R_{ex}$ and $R_{ix}$ are the aggregate excitatory and inhibitory firing rates of the local network, and $I_{in}$ is the feedforward input. The firing rate *R* of the output neuron is computed from the synaptic currents with $R = F_e(I_s)$, where $F_e$ is a monotonic increasing function. The model dynamics are given by

$$
\begin{aligned}
\tau\frac{dI_s}{dt} &= -I_s + J_{ee}R_{ex}*G - J_{ie}R_{ix} + J_{inp}S\left(pc\right); R = F_e\left(I_s\right)\\
\tau\frac{dI_{ex}}{dt} &= -I_{ex} + I_{exin}; R_{ex} = F_e\left(I_{ex}\right)\\
\tau\frac{dI_{ix}}{dt} &= -I_{ix} + I_{ixin}; R_{ix} = F_i\left(I_{ix}\right)
\end{aligned}\tag{4}
$$

We used $\tau$ = 15 ms, $J_{ee}$ = 12 nA/sp/s, $J_{ie} = \frac{J_{ee}}{E/Iratio}$ , and $J_{inp}$ = 12.5 nA/sp/s. The *E/I ratio* was varied between 1.75 and 2.25. The gain of the excitatory current, *G*, varied between 1 and 55. The stimulus *S* varied logarithmically with the percent stimulus contrast, *pc*,

$$S\left(pc\right) = \left(15 + 25 * log_{10}\left(pc\right)/65\right),\tag{5}$$

where 2.5 %<*pc*<100 %. The stimulus was presented for 300 ms. The reported rates are steady-state rates, sampled at 150 ms. Optogenetic suppression was modeled as a reduction in $R_{ex}$ of 12.9 sp/s (in accordance with experimental data). For excitatory neurons, the firing rate *R* was computed from the synaptic current $I_s$ as $R = F_e(I_s)$:

$$F_e\left(x\right) = 100/\left(1 + e^{-0.1\left(x-50\right)}\right),\tag{6}$$

while for inhibitory neurons we used function $F_i$:

$$F_i(x) = 150/\left(1 + e^{-0.13(x-46)}\right). \tag{7}$$

By changing the local excitatory network gain $G$ (in *Equation 3*) we were able to reproduce all four response types found in the experiment (see *Figure 4*). We examined whether changing the sensitivity of the feedforward input, rather than the local network input, could produce similar effects as altering the gain of the local excitatory network. To this end, we fixed the local excitatory network gain $G$ to 1 and multiplied the feedforward drive (*equation 5*) by a gain factor that varied between 0.01 and 55. This arrangement, however, could not replicate all four contrast response types observed experimentally (see also *Figure 3—figure supplement 2*).

## Acknowledgements

The authors wish to thank Neal Waxham for kindly providing the monoclonal mouse CamKIIα antibody. This work was funded by NIH grants U01MH109146 (VD) and R01GM027750 (JLS).

## Additional information

### Funding

| Funder | Grant reference number | Author |
|---|---|---|
| National Institutes of Health | 5U01MH109146 | Valentin Dragoi |
| National Institutes of Health | R01GM027750 | John L Spudich |

The funders had no role in study design, data collection and interpretation, or the decision to submit the work for publication.

### Author contributions

Ariana R Andrei, Data curation, Formal analysis, Investigation, Software, Validation, Visualization, Writing - original draft; Samantha Debes, Xiaoqin Liu, Elsa Rodarte, Investigation, Methodology; Mircea Chelaru, Methodology, Software; John L Spudich, Methodology, Validation; Roger Janz, Data curation, Methodology; Valentin Dragoi, Conceptualization, Funding acquisition, Investigation, Project administration, Resources, Supervision, Writing - review and editing

### Author ORCIDs

Ariana R Andrei (iD) http://orcid.org/0000-0003-2152-2580
John L Spudich (iD) http://orcid.org/0000-0003-4167-8590
Valentin Dragoi (iD) http://orcid.org/0000-0002-9526-0926

### Ethics

This study was performed in strict accordance with the recommendations in the Guide for the Care and Use of Laboratory Animals of the National Institutes of Health. All of the animals were handled according to approved institutional animal care and use committee (IACUC) protocols of the University of Texas, Houston. The protocol was approved by the Committee on the Ethics of Animal Experiments of the University of Houston (Protocol number: AWC-20-0075). All surgery was performed under general anesthesia, and every effort was made to minimize suffering.

### Decision letter and Author response

Decision letter https://doi.org/10.7554/eLife.66400.sa1
Author response https://doi.org/10.7554/eLife.66400.sa2

## Additional files

### Supplementary files
• Transparent reporting form
• Source data 1. *Figure 1*, *Figure 2*, *Figure 3* and *Figure 5* data.

### Data availability
All data generated or analyzed during this study are included in the manuscript and supporting files. Source data files are provided.

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
