## [Decision Letter]

**Acceptance summary:**

The visual cortex contains an abundant of recurrent connections that are critical for computation. The authors show how local optogenetic inactivation of a column in macaque V1 leads to heterogeneous and layer-dependent activity changes mediated by lateral interactions. These changes range from full suppression to facilitation, and a mixture of both. The authors further demonstrate that these lateral interactions determine behavioral outcomes, hence suggesting that behavioral outcomes cannot be predicted based on the focal inactivation alone.

**Decision letter after peer review:**

Thank you for sending your article entitled "Heterogeneous side-effects of cortical inactivation in behaving animals" for peer review at *eLife*. Your article has been reviewed by four peer reviewers, including Martin Vinck as the Reviewing Editor and Reviewer #1, and the evaluation has been overseen by Joshua Gold as the Senior Editor.

As you will see from the reviewers, they are mixed, although three reviewers make numerous positive remarks (and also some critical comments) about the interest of the paper. Notably there are a number of concerns that need to be addressed, in particular the potential need for additional experiments related to the sampling of the contrast. Reviewers raised the concern that sampling of the contrast axis should be based on 7-8 contrasts, and that the under-sampling of contrast space in combination with low trial numbers might affect the claims about increased activity at low vs. high contrasts and the conclusions on heterogeneity. It would be important to understand how this concern will be addressed. One option could be, e.g., to validate the findings in a smaller set of experiments, if possible. Besides this point, you will find numerous other concerns to be addressed below.

Essential revisions:

*Reviewer #1:*

The authors use focal optogenetic inactivation together with dual laminar recordings in awake primates to study the impact of activity in one column on the activity in another column. With this technique, they investigate the nature of lateral interactions and normalization mechanisms in V1. The key findings are that: (1) Optogenetic inactivation does not yield an overall facilitation or suppression in a nearby column. (2) This is explained by heterogeneous changes in activity that are stimulus dependent, with four main types of responses, the most common one being general suppression across all stimulus contrasts, in particular in the supra-granular layers. (3) A simple E/I network can predict these four types depending on two parameters. (4) There exists a high degree of variability across sessions, however in sessions with predominantly Type-1 responses (suppression), behavioral outcome is also affected. Together, these findings suggest that the effect of optogenetic inactivation on the local circuit is highly complex, that lateral interactions are highly heterogeneous across cells and that focal optogenetics inactivation might impact behavior in a complex manner. My main comments pertain to the statistical analyses:

1. It was not entirely clear to me how the four subgroups were statistically quantified.

A. Where the individual neurons that belong to the four types statistically significant?

B. Was the distribution in four types statistically different from chance?

C. How was the choice of four categories motivated, was this done through a clustering mechanism?

2. It would be useful to discuss how the contrast-dependence of lateral interactions (with visual stimuli) fits or does not fit with the present results.

3. The nature of the model is not entirely clear: Which variables represent the local site and which variables represent the distant site? Do both have E/I neurons? What are the connections between those?

4. It would be useful to explain better what standard normalization models would predict. There is some discussion on this, but it is not clear why one would expect suppression of a distal site rather than activation at these retinotopic distances. In this context it would also be useful to discuss Mexican-hat profiles of activation/suppression in relationship to the present findings.

5. Detailing criteria for spike sorting would be useful for future studies comparing to the present findings.

6. Figure 2C misses a color bar.

7. Unless my PDF renderer has a problem, Figure S5 seems to be the wrong figure (duplicate of Figure 5) and this control should be fixed.

*Reviewer #2:*

Andrei and colleagues performed an optogenetic experiment in area V1 of monkeys, whereby they inactivated glutamatergic neurons using a lentiviral vector approach with CamKII as promotor and the chloride conducting GtACR2 opsin. To investigate the local network effects of the optogenetic inactivation, they recorded single and multiunit activity nearby the optic fiber and at about 300 micron distance. The major claim of the authors is that local reversible inactivation leads to unexpected or unpredictable activity changes in (nearby and) distant neurons. Neurons affected by the light were classified in 4 groups, depending on the optogenetic-induced dynamic activity changes. These 4 types of responses could be predicted based on a simple spiking model with a linear combination of excitatory feedforward drive and local recurrent excitatory and inhibitory inputs. In addition, monkeys also performed a contrast detection task, and performance was impaired only when the activity in the majority of the (recorded) neurons was suppressed.

This study yielded a number of interesting findings, but contrary to the framing of the authors, ("unexpected" "unpredicted" "most surprising" "unpredictable", "off-target effects have never been investigated…"etc.), a variety of immediate downstream "off-target" effects after optogenetic activation and inactivation have been amply described in primates -already starting with the first optogenetic study in monkeys (Han et al., 2009). The main 'selling' point of the study is unsurprising. Virtually every study so far showed, predictably, a mixture of facilitation and suppression at single unit level, independent of the type of opsin used. In general, more suppression is found with a hyperpolarizing opsin and more (or net) enhancement with depolarizing opsins. That said, the current findings whereby neighboring and distant effects at neuronal level are compared, are certainly a nice addition to the literature, as is the apparent division in four response classes. The authors should significantly tone down their language, however. Of course, this has also consequences for the impact of the paper.

This reviewer also questions the usefulness of GtARC2 due the exceedingly long after hyperpolarization, which may have contributed to the potentially stronger off-target (i.e. facilitation) effects compared to other hyperpolarizing opsins. Also, it is unfortunate that continuous stimulation was used, which prohibits a latency analysis in the distant neurons. Such an analysis would have made the discussion about direct versus indirect effects much simpler, at least compared to the current argumentation of the authors.

The authors argue twice that it is troublesome to interpret optogenetic -induced effects without measuring neuronal activity at distant sites during light exposure. This reviewer does not argue about the usefulness of concurrent recordings. However, it is almost impossible to cover all sites which may show off-target effects. In the present study, one only recorded at (some sites) at 300 micron distance. Yet, all sites connected with the neurons directly affected by the light might show "off-target" effects, even those in remote areas. It is impossible to 'cover' all these sites with electrophysiological recordings. Hence the solution offered by the authors is not workable. Whole brain imaging may be an alternative, yet it remains challenging to relate changes in imaging signals with alterations in neuronal activity.

The authors categorized the optogenetic induced neuronal responses at a distance in four classes, which is a nice finding. It is unclear, however, how the neurons were clustered. Unless I missed it, no clustering approach with objective criteria to determine the number of relevant clusters was used. Please elaborate on this.

The authors emphasize the degree of heterogeneity of the indirect network effects, but this may be highly related to the transduction efficiency and layer-specificity of transduction.

Figure S5 is the same as Figure 5.

*Reviewer #3:*

The manuscript investigates the important question how local optogenetic silencing of V1 neurons affects neural populations that are located nearby, and whether these effects can impact on behaviour. Based on contrast response functions, the authors identify 4 different effect types, which range from inhibition only through mixed effects that depend on stimulus contrast to facilitation only.

The diversity of effects could be replicated in a network model by varying stimulus sensitivity of network inputs.

Behavioural effects on contrast detection occurred only when off target effects were dominated by type I effects, i.e. those that show inhibition at all contrast levels.

The idea of studying the effects of optogenetic silencing on off target neurons is important, but there are problems with the identification of different effect types given the coarse sampling of stimulus contrasts. Also, the effect on behaviour would benefit from more detailed investigation, e.g. limiting the stimulus dimensions to off target locations, as well as obtaining a lager data base to study effects of other response types on behaviour more quantitatively.

1. In methods the injection procedures are described, but it remains unclear how 'the deepest point' of a column was determined? Was it a fixed depth that was targeted? Also, the 5 injections per column were probably evenly spaced. But the distance between sites should still be given.

2. Methods: "If either the transient or sustained response during optical stimulation was significantly different (P<0.05, Wilcoxon rank sum test)" -- was this corrected for multiple comparisons? After all 2 tests are done on the same activity.

3. The contrast fitting function (aka Nassi) is problematic due to the number of contrasts tested in a single experiment. It has 4 free parameters for 5 data points (when 4 +0% contrast was used). Also, the equation has no numbering.

4. Were the oriented gratings really flashed at 30Hz? i.e. 33ms on time per stimulus? This could induce strong masking effects.

5. In methods the authors state that the light guide was close to the recording electrode in 7 sessions. In figure 1E it states that n=48, while in the text it states that n=41 when the statistics are mentioned. Also, it is unclear whether this is neurons or contacts? Were there depth differences for these n=48(41).

6. Were the effects of light stimulation at different contrasts corrected for multiple comparison? If not, they should be.

7. Unless I am mistaken, equation 1 has 4 free parameters (r0, rmax, P,Q, with c50 and n fixed) that are fit to each neuron with optogenetic stimulation? If so, this is problematic for the reasons mentioned above. The authors only measured 5 data points, and the model is thus likely to overfit. Hence the variance accounted for is not very impressive. If my understanding of the fitting procedure is mistaken, then the authors should explain in detail how it was implemented.

8. The authors should describe in detail how the classification of the 4 response types was arrived at. How was the 'Type' category defined? Ideally this would be done, based on the terms P and Q yielding significant improvements to the fit?

9. The data shown in figure 3D-G are puzzling. The P values are mostly negative, i.e. they seem subtractive, rather than additive? That suggests the network does not provide excitation, unlike stated in the main text. Also, the c50 values of many neurons appear very high, and are in a range where sampling was basically absent. All examples shown in figure 2 have c50 values much lower.

10. While the claim that type 2 neurons are more sensitive to low contrast than other types, is correct, this cannot be inferred from the slope of the fitting function, but from the c50. If a neuron did not respond to any contrast including 20%, but strongly to 100% stimuli, it would have a c50>20%, but could have a very steep response function, which would be an artefact of the fitting in conjunction with the sampling.

*Reviewer #4:*

In this study the authors examine the effects of optogenetic inactivation directed onto a cortical site on neuronal activity of nearby lateral loci (off-target), and on behavioral performance.

They find that inactivation of the superficial layers, while reliably suppressing activity at the inactivation site, causes heterogeneous effects at off-target cortical loci, with some laminar-bias. The authors further determined that changes in behavioral performance were consistently observed only in sessions where suppressive effects among recorded cells dominated. The study is overall well executed, and well presented, and it is of interest to a broad audience. Some additional analysis would strengthen the claim that photostimulation at the inactivation site does not spread to nearby loci, where the recordings are made. Moreover, it is unclear whether the observed heterogeneous effects would become more homogenous at higher light intensities.

1) Could the effects seen at off-target sites depend on the specific temporal dynamics of the inactivating opsin used in this study? This point should be addressed in the discussion.

2) To strengthen the main claim of the paper, that off-target sites 300µm away from the inactivation site are not directly inactivated by light, the authors should determine the onset latency of suppressive effects at the on vs off-sites. The expectation would be that the latter are suppressed significantly later in time than the former if, indeed suppression is a network effect. While I do not expect that light could inactivate directly the deep layers at the off-site, it could easily spread 300µm away and directly affect the superficial layer cells. In support of this, the type I cells dominate in the superficial layers. Moreover, some of the off-site suppression, e.g. in the example cells in Figure 1F-G, seems to occur very early, possibly suggesting direct inactivation by light spreading to the off-site. Importantly, the latency analysis should be performed on a layer-by-layer basis because it is possible that only the superficial layer cells at the off-site are directly affected by light, while those in deep layers are a result of network effects. On p. 4 the authors state that:" The lack of suppression in the blank condition indicates these responses cannot be due to direct activation of GtACR2 at the distal site". First, the authors could strengthen this claim by showing this at the population level, rather than simply showing the few example cells in Figure 1F-I. On p. 6 the authors further make the point that the temporal profile of suppression seen at the inactivation site is rather different from that at the off-site, with long-lasting responses only found at the inactivation site near the light source. However, one could imagine that these different profiles may result from different light intensities at the inactivation site compared to the off-site; light scatter in tissue may result in reduced photostimulation intensity at the off-site. For the same reason, reduced light intensity at the off-site may also cause significant suppression only when the cell is driven by a visual stimulus, but not be apparent in the baseline response. For all these reasons, an analysis of onset latency of suppression (in the stimulus-driven condition) at the inactivated site vs the off-site could strengthen the authors' claim that the off-site is not directly affected by light.

3) I could not find the light intensity values used for inactivation anywhere in the manuscript. This should be added.

4) Figure 1E. rather than one example cell, it would be preferable to show the full laminar profile of suppression at the photoactivated site to demonstrate that light is, indeed, limited to the SG layers. This is shown in Figure S1B-C, but this figure is difficult to interpret correctly because the Y axis is not labeled, and the estimated top and bottom of cortex as well as L4C are not indicated on the laminar plot.

5) Figure 2E. Judging from the CSD analysis, here the top of the cortex would seem to be contact 2, rather than 0, and the thickness assigned to the G layer is too large. The latter in vivo typically spans about 3, not 4, contacts (if the penetration are vertical which this appears to be). Moreover, the earliest current sink, which is the criterion for identifying layer 4C, would seem such that the top of G should be moved down by at least one contact. The selection should be based on a latency analysis. I am raising this issue because with layers more properly assigned the laminar data could potentially clean up and appear less heterogeneous.

6) Model. Could the 4 different types of responses depend on intrinsic properties of cells (for example their contrast response function), rather than, or in addition to the lateral network connectivity?

7) Model. What determines how strongly the network is driven by the stimulus, in the model and possibly in the real brain? In other words, what determines network sensitivity? Is this the weights of the local connections?

8) There should be a discussion of whether these results could change depending on photostimulation intensity. Is it possible that at higher light intensities the off-site would be more homogeneously suppressed for ex.?

9) Please add all sample size to the figure legends. For ex, sample sizes are missing in panels C and F of Figure 5.

10) Isn't it odd that effects on behavioral performance in Figure S7E are only seen at 20% contrast given that type 2 cells are suppressed at contrasts {greater than or equal to} 10?

11) P. 25 Layer Identification. The granular layer in the CSD analysis should be defined as the location of the earliest current sink, not the "maximum sink" as sated.

---

## [Author Response]

First, thank you very much for your very useful comments and suggestions. In this revised manuscript we present the results of the analyses that you have recommended and describe the new control experiments we have performed. Our new findings offer additional insight into the response types revealed by cortical suppression and provide additional, strong support for our previous conclusions.

1. The most common question among reviewers was how the 4 response categories were identified, and whether a clustering method was used.Reviewer 1: “How was the choice of four categories motivated, was this done through a clustering mechanism?”; Reviewer 2: “The authors categorized the optogenetic induced neuronal responses at a distance in four classes, which is a nice finding. It is unclear, however, how the neurons were clustered.”; Reviewer 3: “The authors should describe in detail how the classification of the 4 response types was arrived at. How was the 'Type' category defined?”

a. We took several steps to address this point (as described below).

b. We thank the reviewers and editors for suggesting the UMAP clustering algorithm. We have included the results of the UMAP clustering algorithm performed on the firing rates of the different response types, using the correctly ascribed labels (Figure 2 – supplementary figure 1D) and randomized labels (Figure 2 – supplementary figure 1E). UMAP produces 4 clear, non-overlapping clusters when firing rates across contrasts are correctly labeled with the response type, but yields only one cluster when labels are randomly assigned to the same firing rate data. This analysis demonstrates that the classification of response types based primarily on the normalization fits corresponds to differences in firing rates for individual cells and is not due to errors in fitting accuracy. This new figure has been included in the revised Figure 2 – supplementary figure 1 (page 32).

c. As suggested by the editors and reviewers, we also implemented a similar clustering analysis using the t-SNE (t-distribution stochastic neighbor embedding) algorithm, using the identical data as for the UMAP analysis above. The t-SNE algorithm produced similar results, producing better grouping of data with correct labels versus randomized labels (Author response image 1) . However, this method was not as effective compared to the UMAP method and given its redundancy we did not include this analysis in the revised manuscript.

**Author response image 1. sa2fig1:** t-SNE results clustering firing rate differences (laser minus control) across all contrasts for all cells, labeled with ascribed labels (A), or with randomly ordered labels (B). Error ellipses represent the covariance matrix and are centered on the median of tSNE results for each type.

d. Similar to the above clustering analysis above, in order to ensure that the response types categorized using the normalization fits corresponded to the underlying firing rate changes we implemented a classifier. The classifier was trained on the differences in firing rates (laser minus control) across all contrasts for each cell using either the correct labels (Figure 2 – supplementary figure 1F, left side black bar ‘Actual’) or randomly shuffled labels (Figure 2 – supplementary figure 1F, left side gray bar ‘Chance’). The classifier was validated using 25% holdout validation. This procedure was repeated 5 times and the mean ± s.e.m. classifier performance is shown in Figure 2 – supplementary figure 1F. The overall accuracy of the linear discriminant classifier to correctly classify firing rates changes into the ascribed classes was 74.3% ±2.6 (mean ± s.e.m.) compared to 33.2% ±1.9 on the randomized data. For individual response types (Figure 2 – supplementary figure 1F, right side), the area under the receiver operating characteristics curves (‘AUC’) are all significantly above chance (colored versus gray bars), and all AUC>0.8, indicating the classifier performed well for all types. This figure is included in Figure 2 – supplementary figure 1F, and is mentioned in the text (page 6, paragraph 2).

e. As suggested, we have now included a new paragraph to address the classification as a separate issue and describes the methods in detail (page 6, paragraph 2).

f. Taken together, the results of the clustering analysis and the classifier performance convincingly demonstrate that the response types can be well separated based on the measured light-evoked changes in firing rate across contrasts. Thus the identification of 4 response classes do not rely solely on the normalization fits, nor can they be attributable to chance.

2. Reviewers 1 and 3 ask about statistical testing to ensure that the responses are unlikely to be due to chance.

a. To address whether the heterogeneity could arise by chance, we re-examined the MATLAB code where the laser and stimulus responsiveness is statistically tested for each unit and found a typographical error in the methods describing this process. The laser responsiveness of each cell was based on an α value for the Wilcoxon ranked sum test of P<0.005 (0.05 divided by 10 comparisons), not 0.05 as previously stated. To assess responsiveness to the visual stimulus we used an α value of P<0.05, since only one comparison was made for each cell (highest contrast versus 0% contrast). This has now been corrected in the revised methods section (page 27, paragraph 2). Since this method includes a stringent Bonferroni correction for multiple comparisons to identify laser responsive units, this should provide a high degree of confidence that the laser responses are not due to chance.

b. In addition to showing example cells for each response type in Figure 2A, we have also included the population firing rates for each response type in Figure 2B (page 8), showing statistics with multiple comparisons corrections. This figure was previously provided in Supplementary Figure S2.

3. Reviewers 2 and 4 raised the important point of distinguishing direct activity suppression from indirect, network-based suppression. Reviewer 4 suggested that we perform a latency analysis to differentiate direct light suppressive effects from indirect ‘Type 1’ suppressive effects.

As suggested by Reviewer #4 – to test whether the Type 1 (all suppressive) responses at the distal/indirect site could actually be due direct activation of GtACR2 by a small amount of light scattering, we performed a suppression latency analysis. That is, we compared the time required to reach the maximum suppression following the onset of both the visual stimulus and the light for cells at the direct site and for Type 1 cells found at the distal site (Figure 1 – supplementary figure 1L). We also separated the indirect Type 1 cells across layers (Figure 1 – supplementary figure 1M). As expected, we found that the suppression latency at the direct sites was significantly shorter than that at the indirect sites (31.2 ± 1.15ms versus 92.9 ±1.10ms, respectively; P=0.00074 Wilcoxon ranked sum test). Suppression at direct site was also significantly different compared to Type 1 cells across all layers (P=0.0022 ANOVA test, d.f.=3), but no differences were found across layers. This analysis supports our previous interpretation that Type1 cells at the indirect site are suppressed due to their synaptic connections with the directly suppressed cells. This is now addressed in the manuscript (page 4, paragraph 3).

4. Reviewer 3 and the editors raised concerns about the number of contrasts used to fit the contrast response functions and suggested that we perform additional experiments using 7-8 stimulus contrasts.

a. We performed additional experiments using 9 stimulus contrasts to evaluate the accuracy of our previous fitting procedure (Figure 2 – supplementary figure 2). The responses of 54 units were either fit (i) using the mean firing rates from all 9 contrasts (0, 3.5, 5, 10, 15, 25, 50, 75, and 100%), or (ii) using a subset of 5 contrasts (0, 5, 10, 25, and 100%), with a similar range as in the original experiments. However, when we compared the fit parameters obtained from the 9-point and 5-point fits, we found that they were highly similar (assessed by a Pearson correlation coefficient). In particular, the more critical parameters of c50 (Figure 2 – supplementary figure 2C) and slope (Figure 2 – supplementary figure 2D, left) both had a Pearson correlation coefficient of 0.89 (Pearson P<0.001). This is now addressed in the manuscript (page 6, paragraph 3).

b. Interestingly, the above analysis also led us to notice that a portion of cells (~30%) do not have saturating responses at high contrasts. For these cells, the Naka-Rushton fitting parameter associated with c50 no longer corresponds with the contrast that elicits 50% of the maximal response, but rather this parameter approaches 100% contrast (Figure 2 – supplementary figure 2E-F). In our previous manuscript we reported the overall mean value of Naka-Rushton c50 parameter without accounting for these cells, which led to an over-estimation of the actual c50 of the population. To correct this, in the revised manuscript we have measured the actual c50 from the fits (new Figure 3G) and moved the report of the Naka-Rushton fit parameter c50 to Figure 2 – supplementary figure 2G. We now also report the percentage of cells within each of the 4 types that exhibit this non-saturating behavior (Figure 2 – supplementary figure 2H), and show the distributions of the Naka-Rushton c50 parameters (Figure 2 – supplementary figure 2I). Importantly, please note that for the majority of cells, both the empirically-measured c50 (Figure 2 – supplementary figure 2C, above; 12.6% ± 0.98 contrast, mean ± s.e.m., n=214) and the Naka-Rushton c50 parameter was contrast (Figure 2 – supplementary figure 2I) was less than 20% – which is within the range of contrasts sampled in the original manuscript. This is addressed in the manuscript (page 11, paragraph 4).

c. As a side note, the above analysis made us wonder whether these non-saturating cells maybe *do* actually saturate, but only at very high contrasts that our 9-point fits did not sample. To satisfy curiosity, we recorded an additional population of 28 cells and densely sampled at high contrasts (Figure 2 – supplementary figure 2J-L, above). We found that 21% of cells indeed do not seem to saturate (Figure 2 – supplementary figure 2J). This proportion was similar to that observed in the original data (compare Figure 2 – supplementary figure 2L with Figure 2 – supplementary figure 2I, above), leading us to conclude that the cells in the original data with high (>80%) Naka-Rushton c50 parameters represent a real type of cell behavior, and is not due to under sampling of high contrasts. This is addressed in the legend to Figure 2 – supplementary figure 2 (page 33).

d. The above experiments demonstrate that the range and number of contrasts used in the original experiments are sufficient to capture the contrast response parameters almost as well as it could have been obtained using 9 contrasts. Importantly, this analysis shows that the c50 parameter for the majority of cells falls within the range of contrasts tested in the original experiments. This is discussed in manuscript (page 6, paragraph 3).

5. Reviewer 4: “… and the thickness assigned to the G layer is too large. The latter in vivo typically spans about 3, not 4, contacts (if the penetration are vertical which this appears to be).”

a. In our laminar analysis we employed the same laminar definitions as many other physiology studies (Cox et al., 2019; Dougherty et al., 2019; Van Kerkoerle et al., 2017; Westerberg et al., 2019), with a granular layer depth of ~500 microns, as supported by anatomical studies (Lund, 1973; O’Kusky and Colonnier, 1982; Vanni et al., 2020).

b. However, since there is considerable variability in the thickness of the granular layer across anatomical studies (ranging between 360 – 580 microns from our literature survey), we have redefined granular as spanning 300 microns and replotted the laminar distribution of response types for comparison (Figure 2F, Figure 2—figure supplement 3C). Our previous analysis used a convention of 400 µm. Shifting this boundary by 100 µm (1 contact difference) resulted in layer assignment changes (G to SG) for 16 cells (n=7 Type 1, n=5 Type 2, n=4 Type 3 and n=0 Type 4).

6. Reviewer 1: “The nature of the model is not entirely clear: Which variables represent the local site and which variables represent the distant site? Do both have E/I neurons? What are the connections between those?”

The model output is the firing rate of one neuron at the distant site. The activity of this neuron is driven by stimulus-related activity, as well as by excitatory and inhibitory currents representing the local network. This has been clarified in the manuscript (page 12, paragraph 2), and the model schematic has been modified to show that both the local network and the output neuron receive feedforward input (Figure 4A left, page 13)

7. Reviewer 1: “It would be useful to explain better what standard normalization models would predict. There is some discussion on this, but it is not clear why one would expect suppression of a distal site rather than activation at these retinotopic distances. In this context it would also be useful to discuss Mexican-hat profiles of activation/suppression in relationship to the present findings.”

a. We have included the following paragraphs in the discussion (starting page 17, paragraph 3):

“Heterogeneity of normalization has not been well studied, nor is it well-defined for populations of neurons. The denominator of the normalization equation represents the cumulative activity of the local network (or ‘normalization pool’). […] It is quite possible that some of the effects recorded at the 300 µm distance, could be due to activity changes that occurred at longer distance (>500 µm) primarily mediated by long-range inhibitory projections.”

8. Reviewer 2: “This study yielded a number of interesting findings, but contrary to the framing of the authors, ("unexpected" "unpredicted" "most surprising" "unpredictable", "off-target effects have never been investigated…"etc.), a variety of immediate downstream "off-target" effects after optogenetic activation and inactivation have been amply described in primates -already starting with the first optogenetic study in monkeys (Han et al., 2009). The main 'selling' point of the study is unsurprising.”

a. We have noted that previously reported paradoxical effects (i.e. Nassi et al. 2015) were noted for cells recorded within the area where light was applied (page 17, paragraph 3). The novelty of our study is that our heterogeneous responses are away from the light source, in the local network where recordings are seldom made.

b. We have added to the discussion a useful counterpoint from Li et al. (Li et al., 2019), that measures the effects of optical suppression at various distances from the light source, in the mouse (page 19, paragraph 3). This study shows that suppression, in the absence of a sensory stimulus, is localized to an area about the size of the fiber optic. Li et al. (2019) conclude that optical suppression is highly localized to the area of the light, with virtually no lateral effects. The novelty of our study is that we show that once the network is driven by a stimulus, focal suppression produces unpredictable activity ripples in the local network, which has a behavioral impact. Our study is particularly useful for future implementations of optogenetics in NHPs aiming to drive/modulate behavior, which has proven notoriously difficult.

9. Reviewer 3 “The data shown in figure 3D-G are puzzling. The P values are mostly negative, i.e. they seem subtractive, rather than additive? That suggests the network does not provide excitation, unlike stated in the main text? Also, the c50 values of many neurons appear very high, and are in a range where sampling was basically absent. All examples shown in figure 2 have c50 values much lower.”

a. Negative P parameter values are not surprising since in this experiment the local network is suppressed, and the reported P parameters are from laser trials. The actual effect of the network under normal conditions can be inferred to be the opposite of this. We have emphasized that the P and Q parameters reported here were from laser trials (page 11, paragraph 4).

b. The c50 values measured from the fits (rather than the Naka-Rushton c50 parameter) are now shown in Figure 3G. See also point 4C above.

10. Reviewer 4: “Figure 1E. rather than one example cell, it would be preferable to show the full laminar profile of suppression at the photoactivated site to demonstrate that light is, indeed, limited to the SG layers. This is shown in Figure S1B-C, but this figure is difficult to interpret correctly because the Y axis is not labeled, and the estimated top and bottom of cortex as well as L4C are not indicated on the laminar plot.”

Figure 1E is actually a population average of all directly suppressed neurons. Unfortunately we could not obtain clear laminar information for the session in Figure 1-supplementary figure 1 (formerly Figure S1), possibly owing to the presence of 2 probes. However, we did measure the vertical distance between cells that were directly suppressed by light based on the distance between recording contacts (spaced every 100 µm). We found that these directly suppressed cells were clustered within 457.1 ± 104.3 µm (mean ± s.e.m., n=31 cells with maximum suppression within 30 ms of light onset). We have included this information in the revised manuscript (page 26, paragraph 3).

11. Reviewer 4: “Isn't it odd that effects on behavioral performance in Figure S7E are only seen at 20% contrast given that type 2 cells are suppressed at contrasts {greater than or equal to} 10?”

Overall, the Type 2 cells show greater suppression for higher contrasts. This is most obvious in Figure 2A-B above, where the suppression for the 100% contrast is stronger compared to control than the degree of suppression at 10% contrast. It is likely that the 20% contrast was high enough to sufficiently induce suppression in Type 2 cells, while being low enough to observe behavioral changes.

References

Ben-Yishai R, Lev Bar-Or R, Sompolinsky H. 1995. Theory of orientation tuning in visual cortex. *Proc Natl Acad Sci U S A* 92:3844–3848. doi:10.1073/pnas.92.9.3844

Cox MA, Dougherty K, Adams GK, Reavis EA, Westerberg JA, Moore BS, Leopold DA, Maier A. 2019. Spiking Suppression Precedes Cued Attentional Enhancement of Neural Responses in Primary Visual Cortex. *Cereb Cortex* 29:77–90. doi:10.1093/cercor/bhx305

Dougherty K, Cox MA, Westerberg JA, Maier A. 2019. Binocular Modulation of Monocular V1 Neurons. *Curr Biol* 29:381-391.e4. doi:10.1016/j.cub.2018.12.004

Lund JS. 1973. Organization of neurons in the visual cortex, area 17, of the monkey (Macaca mulatta). *J Comp Neurol* 147:455–495. doi:10.1002/cne.901470404

O’Kusky J, Colonnier M. 1982. A laminar analysis of the number of neurons, glia, and synapses in the visual cortex (area 17) of adult macaque monkeys. *J Comp Neurol* 210:278–290. doi:10.1002/cne.902100307

Spiridon M, Gerstner W. 2001. Effect of lateral connections on the accuracy of the population code for a network of spiking neurons. *Netw Comput Neural Syst* 12:409–421. doi:10.1080/net.12.4.409.421

Van Kerkoerle T, Self MW, Roelfsema PR. 2017. Layer-specificity in the effects of attention and working memory on activity in primary visual cortex. *Nat Commun* 8:13804. doi:10.1038/ncomms13804

Vanni S, Hokkanen H, Werner F, Angelucci A, Helsinki B. 2020. Anatomy and Physiology of Macaque Visual Cortical Areas V1, V2, and V5/MT: Bases for Biologically Realistic Models. *Cereb Cortex* 30:3483–3517. doi:10.1093/cercor/bhz322

Westerberg JA, Cox MA, Dougherty K, Maier A. 2019. V1 microcircuit dynamics: Altered signal propagation suggests intracortical origins for adaptation in response to visual repetition. *J Neurophysiol* 121:1938–1952. doi:10.1152/jn.00113.2019